# Entropy-informed Decoding: Adaptive Information-Driven Branching

**Benjamin Patrick Evans** [1]  **Sumitra Ganesh** [2]  **Leo Ardon** [1]

## Abstract

Large language models (LLMs) achieve remarkable generative performance, yet their output quality is dependent on the decoding strategy. While sampling-based methods (e.g., top-$k$, nucleus) and search-and-select based methods (e.g., beam search, best-of-$n$, majority voting) can improve upon greedy decoding, both approaches suffer from limitations: sampling generally commits to a single path, while search often expends excessive computation regardless of task complexity. To address these, we introduce **Entropy-informed DEcodiNg** (EDEN), a plug-and-play, model-agnostic decoding framework that adaptively allocates computation based on the model's own uncertainty, approximating higher-width beam search with *fewer expansions*. At each generation step, EDEN estimates the entropy of the output token distribution and adjusts the branching factor monotonically with the entropy, expanding more candidates in high-entropy regions and following a greedier path in low-entropy regions, improving token efficiency. Experiments across complex tasks, including mathematical reasoning, code generation, and scientific questions, demonstrate that EDEN consistently improves output quality over existing decoding strategies, achieving better accuracy-expansion trade-offs than fixed-width beam search. By treating next-token selection as a noisy maximisation problem, we prove that branching factors monotone in entropy are guaranteed to find better (i.e. more probable) continuations than any fixed branching factor within the same total expansion budget, and derive explicit regret rates characterising the benefit of the adaptive allocation.

[1]JP Morgan AI Research, London, UK [2]JP Morgan AI Research, New York, USA. Correspondence to: Benjamin Patrick Evans <benjamin.x.evans@jpmorgan.com>.

*Proceedings of the $43^{rd}$ International Conference on Machine Learning*, Seoul, South Korea. PMLR 306, 2026. Copyright 2026 by the author(s).

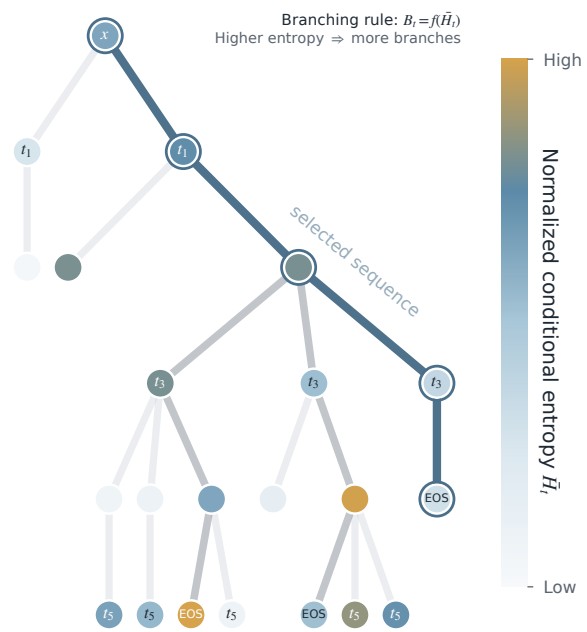

*Figure 1.* **Entropy-informed decoding (EDEN) adaptively allocates search compute.** At each decoding step, EDEN estimates the normalized entropy of the next-token distribution and converts it into a branching factor. Low-entropy states are decoded nearly greedily, while high-entropy states trigger additional exploration. This concentrates computation on uncertain reasoning forks while preserving a high-scoring completed sequence.

## 1. Introduction

Large language models (LLMs) have demonstrated impressive generative capabilities, but the quality of generated outputs depends on the decoding strategy. The simplest approach, greedy decoding, which selects the highest-probability token at each step, is well known to be suboptimal. Greedy decoding can lead to repetitive or incoherent text by making locally optimal choices that sacrifice better global continuations (Holtzman et al., 2020). Even a single low-probability token choice may remove promising continuations, effectively "trapping" the model in erroneous reasoning paths. This issue is especially important in complex, multi-step reasoning tasks. To address the limitations of greedy decoding, various alternative decoding techniques

have been proposed. Sampling-based methods such as *top-k* (Noarov et al., 2026), *nucleus (top-p)* (Holtzman et al., 2020), and *min-p* sampling (Minh et al., 2025) introduce randomness to increase output diversity. For instance, top-$k$ sampling restricts token choices to the highest-probability $k$ tokens, often producing more natural text than greedy sampling. Nucleus sampling truncates the distribution to a cumulative probability mass $p$, sampling only from the most likely tokens. Search-and-select based methods like *beam search*, best-of-$n$, and majority voting, explicitly explore multiple outputs, ultimately choosing the best of the resulting candidates. However, existing approaches suffer from important limitations. Sampling methods typically use fixed hyperparameters ($k$ or $p$) and greedily sample from this reduced set of tokens, potentially overlooking valid lower-probability continuations. Search-and-select based methods, by contrast, often require substantial compute regardless of task complexity, failing to allocate resources *adaptively*. For example, a simple query may not require a full search, yet beam search/best-of-$n$/majority voting still generate many expansions irrespective of this complexity. In essence, sampling methods may explore too narrowly, while search-based decoders lack information-driven computation allocation during their expansions.

In this work, we propose **EDEN** (Entropy-informed DEcodiNg), an *entropy-aware decoding* strategy that dynamically allocates computational effort based on the model's own uncertainty, approximating higher width beam search with fewer expansions. At each generation step, we estimate the entropy of the output token distribution to determine areas to focus compute. This is motivated by recent evidence that entropy captures meaningful token-level uncertainty, calibration error, and reasoning forks in LLM generation (Cao et al., 2026; Wang et al., 2026). High-entropy steps trigger higher branching factors with multiple node expansions, while low-entropy steps follow greedier paths, concentrating compute on the most ambiguous points. Crucially, EDEN leverages the model's own logits without requiring extra training, heuristics, or external rewards. EDEN is fully plug-and-play and compatible with existing base decoding strategies, including sampling-based methods, and works seamlessly even with closed source models where we provide theoretical bounds on the number of samples required, and experiments under limited API access.

Our main contributions are:

- **Adaptive branching.** We propose a novel search-based approach that adapts the computation allocation (in the form of branching factor) based on the estimation of entropy of the model outputs, efficiently finding more likely continuations without exhaustive search.

- **Theoretical grounding and regret rates.** While branching decisions in existing approaches are rela-

tively ad hoc, we prove that branching factors monotone in entropy improve upon fixed branching under mild assumptions. We further derive an explicit regret bound showing that entropy-adaptive compute allocation reduces cumulative decision error relative to uniform fixed branching, yielding sublinear or even bounded regret under appropriate budget scaling.

- **Sampling guarantees for entropy estimation.** For closed source models, we provide theoretical bounds on the number of samples (sublinear) required to estimate the entropy within a given error tolerance, enabling robust branching decisions even when working with limited API access or approximate distributions.

In practice, EDEN provides a principled, adaptive decoding method. Easy tokens are decoded nearly greedily, while ambiguous reasoning forks receive additional exploration. This gives EDEN a favorable accuracy–compute trade-off, especially in multi-step reasoning settings where uncertainty is unevenly distributed across the generation.

## 2. Related Work

Several recent works have incorporated entropy and related information-theoretic measures to guide decoding and control behavior in language models. Simonds (2025) proposes an entropy-aware model-switching strategy, dynamically selecting between large and small LMs based on output entropy to reduce inference cost. Li et al. (2025; 2026) use entropy thresholding to binarily trigger fixed branching (i.e., to branch or not), improving coverage in mathematical reasoning. Entropix (Xjdr-Alt, 2024) incorporates both entropy and the variance of the entropy to adapt decoding strategies, e.g., by inserting chain-of-thought or pause tokens at high-uncertainty points. Rahn et al. (2024) introduce entropy-based activation steering to control LLM agents. Zhang et al. (2025a) propose entropy-based exploration depth, where entropy guides how deeply to search during multi-step reasoning. Potraghloo et al. (2026) propose Top-$H$ decoding, an entropy-bounded sampling method that adaptively truncates the token distribution before sampling. HARP (Storaï & Hwang, 2025) introduces a hesitation-aware reframed forward pass that uses token-level entropy to detect uncertainty and selectively apply extra computation during inference, demonstrating how entropy-guided adaptive computation can enhance transformer performance without retraining. Han et al. (2024) leverage entropy to determine self-referential insertion statements in long-form text generation. Each of these works begins to show the usefulness of considering the model's entropy; however, none of the existing works consider what we propose: using entropy to dynamically adjust the width of the search branching factor, providing a more principled approach to adapting beam search widths (Yu-Hsiang et al., 2018).

Unlike prior entropy-based approaches that rely primarily on thresholding or ad hoc heuristics, we introduce a provably justified monotone allocation rule that directly ties the branching factor to entropy. Moreover, we provide formal sample-complexity guarantees for entropy estimation, ensuring compatibility even with closed-API models where only limited sampling and/or limited model logits are available.

## 3. Method

EDEN, presented in Algorithm 1 (and visualised in Figure 1), is guided by the intuition that the *shape* of the output distribution provides valuable information about the model's uncertainty. Prior work has observed: *"Factual sentences are likely to contain tokens with higher likelihood and lower entropy, while hallucinations are likely to come from positions with flat probability distributions with high uncertainty."* (Manakul et al., 2023). Motivated by this, we use entropy as a signal to guide adaptive exploration: rather than focusing on a fixed number of expansions at each decoding step, we base the number of branches on the (estimated) entropy of the output distribution. This adaptive branching allows for dynamic allocation of computational effort, focusing more search on uncertain regions, allowing context-aware branching. Intuitively, the proposed approach can be seen as *approximating* a much higher width beam search while requiring significantly fewer expansions by adapting the branching factor based on the output's entropy.

### 3.1. Entropy-guided branching

**Notation** Let $y$ be an output from the vocabulary $\mathcal{V}$. Let $\mathbf{y}_{1:t}$ denote a partial sequence of length $t$, and $T$ be the maximum allowed sequence length. Let the model's predicted next-token distribution at step $t$ be $P(y_{t+1} \mid x, \mathbf{y}_{1:t})$, where $x$ is some context, $\mathbf{y}_{1:t}$ the sequence so far. For shorthand notational convenience, $\boldsymbol{p} = (p_1, \ldots, p_V)$ denote the sorted probabilities so that $p_1 \geq p_2 \geq \cdots$. The (Shannon) entropy of this output is denoted as $H_t(y_{t+1} \mid x, \mathbf{y}_{1:t}) = -\sum_i p_i \log p_i$, and the perplexity $\mathrm{PP}_t = \exp(H_t)$ (note that when the conditions are dropped, it is always assumed to be conditional on the context as described above). A notation table is presented in Appendix A.

Here, we assume we are trying to maximise a score, where we use the cumulative log-probability of the sequence: $s(\mathbf{y}_{1:t}) = \sum_{i=1}^{t} \log P(y_i \mid x, \mathbf{y}_{1:i-1})$ (and discuss alternative scorers in Appendix E.4). To fairly compare sequences of different lengths, we apply a length penalty $\mathrm{Score}_\alpha(\mathbf{y}_{1:t}) = \frac{s(\mathbf{y}_{1:t})}{t^\alpha}$, where $\alpha$ controls the bias toward longer or shorter sequences (Johnson et al., 2017).

---

**Algorithm 1** Entropy-Informed Decoding

1: **Input:** Input $x$, model $M$, max tokens $T$, vocab size $\mathcal{V}$, max beam size $B_{\max}$, length penalty exponent $\alpha$
2: **Output:** Highest scoring decoded sequence $\mathbf{y}_{1:t}$
3: Run greedy decoding to obtain initial lower bound $S^* \leftarrow \mathrm{Score}_\alpha(y^{\mathrm{greedy}})$
4: Initialize search tree with root node $\mathcal{R}$ representing empty sequence $\mathbf{y}_{1:0}$
5: $\mathcal{B} \leftarrow \{\mathcal{R}\}$ {Active beam candidates}
6: **while** $\mathcal{B}$ not empty **do**
7:    $\mathcal{A} \leftarrow$ active nodes in $\mathcal{B}$ not ending in EOS and with $t < T$
8:    **if** $\mathcal{A} = \emptyset$ **then**
9:       **break**
10:    **end if**
11:    Compute next-token distributions: $P(y_{t+1} \mid x, \mathbf{y}_{1:t})$ for all contexts $\mathbf{y}_{1:t} \in \mathcal{A}$
12:    Estimate entropy $\hat{H}_t(\mathbf{y}_{1:t})$ for all $\mathbf{y}_{1:t} \in \mathcal{A}$
13:    Compute adaptive branching factors: $B_t = \max\left(1, \lfloor B_{\max} \cdot \bar{H}_t(\mathbf{y}_{1:t}) \rfloor\right)$ for all $\mathbf{y}_{1:t} \in \mathcal{A}$
14:    **for** each $\mathbf{y}_{1:t} \in \mathcal{A}$ **do**
15:       Select top-$B_t$ tokens from $P(y_{t+1} \mid x, \mathbf{y}_{1:t})$: $\{y_{t+1}^{(j)}\}_{j=1}^{B_t}$
16:       **for** $j = 1$ to $B_t$ **do**
17:          Form candidate: $\mathbf{y}_{1:t+1} = \mathbf{y}_{1:t} \| y_{t+1}^{(j)}$
18:          Compute cumulative log-prob: $s(\mathbf{y}_{1:t+1}) = s(\mathbf{y}_{1:t}) + \log P(y_{t+1}^{(j)} \mid x, \mathbf{y}_{1:t})$
19:          Estimate future score (Appendix C.2)
20:          **if** $y_{t+1}^{(j)} = $ EOS **then**
21:             $\overline{S}(\mathbf{y}_{1:t+1}) = \underline{S}(\mathbf{y}_{1:t+1}) = \frac{s(\mathbf{y}_{1:t+1})}{(t+1)^\alpha}$
22:          **else**
23:             $\overline{S}(\mathbf{y}_{1:t+1}) = \frac{s(\mathbf{y}_{1:t+1}) + (T-t-1) \cdot \log(1)}{T^\alpha}$
24:             $\underline{S}(\mathbf{y}_{1:t+1}) = \frac{s(\mathbf{y}_{1:t+1}) + (T-t-1) \cdot \log\left(\frac{1}{V}\right)}{T^\alpha}$
25:          **end if**
26:          Add promising candidates *only*
27:          **if** $\overline{S}(\mathbf{y}_{1:t+1}) \geq S^*$ **then**
28:             Add $\mathbf{y}_{1:t+1}$ to candidate pool $\mathcal{B}$
29:             Update $S^* \leftarrow \max(S^*, \underline{S}(\mathbf{y}_{1:t+1}))$
30:          **else**
31:             **break** {Remaining completions are lower ranked}
32:          **end if**
33:       **end for**
34:    **end for**
35:    Retain top-$B_{\max}$ scoring candidates to form next $\mathcal{B}$
36: **end while**
37: **return** Best sequence found with highest normalized score

---

### 3.1.1. WHY ENTROPY

We view the next-token selection at step $t$ as a noisy maximization problem. Let

$$V_t(i) = \log P(i \mid x_t, \cdot) + \text{OPT}_{t+1}(i),$$

$$i^* = \arg\max_i V_t(i), \qquad \Delta_t(i) = V_t(i^*) - V_t(i) \geq 0.$$

where OPT is the (unknown) actual optimal value of the future score. As OPT is unknown, let $\hat{V}_t(i)$ be an estimator of $V_t(i)$ based on $m_t$ units of compute (e.g., rollouts/branches), where $\hat{V}_t(i) - V_t(i)$ has sub-Gaussian noise with variance proxy $\sigma_t^2 = \delta^2/m_t$ (Appendix E.1), a relatively standard estimation assumption; see, e.g., Vershynin (2018) for general properties of sub-Gaussian estimators, and Lattimore & Szepesvári (2020) for the widespread use of sub-Gaussian noise models in score and value estimation. Standard concentration plus a union bound gives the probability of making an estimation error as:

$$\Pr(\text{mistake at } t) = \Pr\left(\hat{V}_t(i^*) < \max_{i \neq i^*} \hat{V}_t(i)\right)$$
$$\lesssim \sum_{i \neq i^*} \exp\left(-c\, m_t\, \Delta_t(i)^2\right).$$

where $c > 0$ is some generic positive constant, which depends only on this variance proxy. Hence, $m_t$ should grow with the *statistical hardness* of step $t$. What we argue here is that **entropy provides a natural, computable proxy for this hardness**. At proof time, we will treat the scalar $m_t$ as the effective sample budget consumed at time $t$. The essential property we use in the proofs is the monotonicity of error with $m_t$, and show that $m_t$ should scale with $H_t$.

To begin, we demonstrate that entropy tells us the number of plausible strong next token candidates (Lemma 3.1). Intuitively, higher entropy means more candidates worth branching on.

**Lemma 3.1** (Entropy controls the number of plausible winners)**.** *For output distribution $p$, the most probable output satisfies $p_1 \geq e^{-H}$. Moreover, for any $\varepsilon \in (0,1)$, the $\varepsilon$-typical set $S_\varepsilon = \{i : p_i \geq \text{PP}^{-1/\varepsilon}\}$ has mass at least $1 - \varepsilon$ and cardinality at least $(1-\varepsilon)\text{PP}^{1/\varepsilon}$.*

*Proof.* For a random index $I \sim p$, define the surprise $X = -\log p_I$. Then $\mathbb{E}[X] = H = \log \text{PP}$. By Markov, $\Pr(X > H/\varepsilon) \leq \varepsilon$, so $\Pr(p_I < \text{PP}^{-1/\varepsilon}) \leq \varepsilon$. Thus, the set $S_\varepsilon$ carries probability mass at least $1 - \varepsilon$. Since each $i \in S_\varepsilon$ contributes at least $\text{PP}^{-1/\varepsilon}$ mass, we must have $|S_\varepsilon| \cdot \text{PP}^{-1/\varepsilon} \geq 1 - \varepsilon$, giving the claimed bound. $\square$

Thus the *effective number of near-optimal candidates* is well captured by $\text{PP}_t$: low $H_t$ implies $\text{PP}_t \approx 1$; high $H_t$ implies $\text{PP}_t \gg 1$.

Intuitively, entropy controls how many next tokens are plausible enough to deserve exploration.

**Lemma 3.2** (Log-gap between top two probabilities)**.** *Let $\gamma = \log p_1 - \log p_2$. Then*

$$\gamma \geq \log\left(\frac{p_1}{1-p_1}\right) \geq \log\left(\frac{e^{-H}}{1-e^{-H}}\right).$$

*Proof.* Since $p_2 \leq 1 - p_1$, $\log p_2 \leq \log(1-p_1)$. Thus $\gamma \geq \log(p_1/(1-p_1))$. By the previous lemma, $p_1 \geq e^{-H}$, yielding the final inequality. $\square$

Consequently, as $H_t \downarrow 0$, $\gamma_t \to \infty$ (easy step); as $H_t \uparrow \log V$, $\gamma_t \to 0$ (hard step). This result complements Lemma 3.2: together, they show that entropy governs both the breadth of plausible candidates and the sharpness of separation between them, justifying its role as a proxy for step difficulty. Thus, entropy is well positioned for determining how many samples we should take, as we show in Proposition 3.3.

To connect entropy to effective score gaps, we assume a mild distributional Lipschitz continuity condition: small perturbations in the next-token distribution induce only bounded changes in continuation values (motivated in Appendix E.3.). This assumption is analogous to the well-known simulation lemma in reinforcement learning (Kearns & Singh, 2002; Szepesvári, 2022) and is widely used in planning and sample-complexity analyses.

**Proposition 3.3** (Required budget increases with entropy)**.** *Under the assumption of distributional Lipschitz continuity: that the continuation value changes smoothly with respect to perturbations in the next-token distribution, we have:*

$$\left|\text{OPT}_{t+1}(i) - \text{OPT}_{t+1}(j)\right| \leq$$
$$\Lambda \cdot d\left(P(\cdot \mid x_t, i, \cdot),\, P(\cdot \mid x_t, j, \cdot)\right) \quad (1)$$

*for some metric $d$ (e.g., total variation), where $\Lambda$ is the Lipschitz constant and can be taken proportional to the remaining horizon $T$ under bounded per-step scores. Define the effective gap*

$$\Delta_t^{\text{eff}} = \min_{i \neq i^*}\left[\log p_1 - \log P(i \mid x_t) - \Lambda\, d_{t+1}(i, i^*)\right].$$

*Then $\Delta_t^{\text{eff}}$ decreases monotonically with $H_t$. To guarantee $\Pr(\text{mistake at } t) \leq \delta_t$, it suffices to take*

$$m_t \gtrsim \frac{1}{\left(\Delta_t^{\text{eff}}\right)^2} \log\frac{\text{PP}_t}{\delta_t}.$$

(see Appendix E.1). Since $\text{PP}_t$ increases and $\Delta_t^{\text{eff}}$ decreases with $H_t$, the required budget $m_t$ is an increasing function of $H_t$, motivating a branching factor that is monotone with entropy.

### 3.1.2. BRANCHING

To convert entropy into a numeric branching factor to guide the decoding, we use a basic piecewise-linear monotonic scaling function $f(H, B_{\max})$ (based on Proposition 3.3):

$$f(H, B_{\max}) = \max\left(1, \lfloor B_{\max} \cdot \bar{H} \rfloor\right),$$

where $B_{\max}$ is a hyperparameter specifying the maximum branching factor allowed (i.e., maximum beam width), and $0 \leq \bar{H} \leq 1$ is the normalized entropy $\bar{H} = \frac{H}{\log \mathcal{V}}$. This function $f$ governs how many outputs to explore from the next-token distribution at each step, justified by Theorem 3.4.

**Theorem 3.4** (Entropy-adaptive branching reduces regret up to constants)**.** *Under the assumptions in Section 3.1.1, there exist constants $a, b > 0$[1] such that choosing*

$$B_t = \max\left\{1, \lfloor a \cdot \phi(H_t) + b \rfloor\right\}$$

*with monotone $\phi$, e.g.,*

$$\phi(H) = \frac{H}{\log \mathcal{V}}$$

*achieves expected cumulative regret $R_t$*

$$\mathbb{E}[R_T] = \sum_{t \leq T} \sum_{i \neq i^*} \Delta_t(i) \Pr(\text{choose } i \text{ at } t)$$

*strictly smaller than that of any fixed-branch policy when the output distributions' entropy varies across decoding steps (Appendix E.2).*

This variation in entropy is typically observed in LLM generation processes (Wang et al., 2026; Cao et al., 2026), further supporting the theorem's relevance to practical decoding.

*Sketch.* Combine Lemmas 3.1–3.2 to express both the candidate count ($\text{PP}_t$) and the effective gap ($\Delta_t^{\text{eff}}$) as monotone functions of $H_t$. Plug these into

$$\Pr(\text{mistake at } t) \lesssim \text{PP}_t \cdot \exp\left(-c\, m_t\, (\Delta_t^{\text{eff}})^2\right).$$

Choosing $m_t \propto \phi(H_t)$ equalizes the exponent across steps up to constants: small $H_t$ needs little compute; large $H_t$ gets more. This lowers cumulative regret relative to any fixed $m_t$. □

**Explicit regret rate.** Under a standard uniform-gap condition, the regret guarantee can be made explicit. Suppose the score estimates are sub-Gaussian, the effective gap is bounded below by $\Delta_{\min} > 0$, the instantaneous regret is bounded by $G$, and the effective candidate set size is

bounded by $P_{\max}$. Then the expected cumulative regret satisfies

$$\mathbb{E}[R_T] \leq G P_{\max} \sum_{t=1}^{T} \exp\left(-c m_t \Delta_{\min}^2\right),$$

for a constant $c > 0$. Thus, with a constant per-step budget $m$, regret scales as

$$O\left(T \exp\left(-cm\Delta_{\min}^2\right)\right),$$

whereas choosing

$$m_t = \Omega\left(\Delta_{\min}^{-2} \log T\right)$$

is sufficient to obtain sublinear or bounded regret depending on the constant. Full details are provided in Appendix E.2.1.

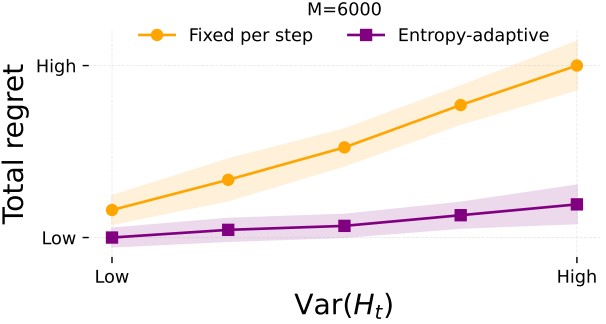

*Figure 2.* Fixed versus adaptive sampling. In the fixed case, every step is allocated the same sampling budget $m_t = \frac{M}{T}$. In the adaptive case, $m_t$ is allocated proportional to entropy $m_t \propto H_t$. The x-axis shows increasing variance $\text{Var}(\mathbf{H}_t)$ of the output distributions. According to Theorem E.2, whenever $\text{Var}(\mathbf{H}_t) > 0$ entropy adaptive allocation dominates fixed allocation (with $M \gg 0$), explaining the improved performance as variability increases.

We demonstrate an example of this adaptive allocation and corresponding reduction in regret with Monte Carlo simulations in Figure 2.

This adaptive branching has the following essential characteristics: If the model is confident, branching is minimal (often just greedy). If the model is uncertain, the algorithm explores multiple alternative continuations. This method is plug-and-play: EDEN can be substituted in place of existing decoding strategies, such as greedy, or integrated on top of existing approaches, such as top-$k$ or nucleus, by modifying the support of the entropy calculation (to consider only the reduced set of tokens rather than the full vocabulary). Additionally, EDEN does not require any model retraining or specialized reward models (although these can be used, Appendix B.2), depending only on the model's own outputs.

## 4. Experiments

We evaluate EDEN on a suite of standard LLM tasks and compare it against widely used decoding baselines. Our

---

[1] In the later sections, for simplicity, we have assumed $a = 1, b = 0$, but such parameters could be tuned for the task at hand.

evaluation focuses on demonstrating the quality and token efficiency of the generations, as well as sensitivity testing key parts of the algorithm.

## 4.1. Configuration

**Model** We use Llama-3.2-3B-Instruct (Touvron et al., 2023) (with default hyperparameters of temperature= 0.6), and provide the same breakdowns in the Appendix B for additional model families, including Gemma3 (Team et al., 2024), IBM Granite (Granite Team, 2024), and base results on Mistral (Jiang et al., 2023). This gives a range of model sizes (from 1B-7B) and model families to ensure performance improvements are robust and model-agnostic. While we only evaluate up to 7B, similar high-entropy points are known to exist in up to at least 70B parameter models (Wang et al., 2026; Cao et al., 2026).

**Datasets** We evaluate across a range of complex tasks, including Math, Programming, and Science. Specifically, GSM8K (Cobbe et al., 2021) and MATH 500 (Lightman et al., 2024) for mathematical reasoning, HumanEval (Chen et al., 2021) for code generation, and SciBench (Wang et al., 2024) for science questions. This gives a broad range of tasks, each with verifiable answers. For each task, we set the maximum generation length $T = 400$, giving ample room for sufficient answers.

**Comparisons** We compare EDEN against commonly utilised search and decoding strategies, including Greedy decoding, Top-$k$ Sampling, Top-$p$ (Nucleus) sampling, Top-$H$ sampling, Min-$p$ Sampling, Majority voting, best-of-$n$, Beam Search, and Diverse Beam Search. We choose these approaches as they have standardized official HuggingFace implementations. For the hyperparameters, we use the ones as specified in the corresponding models' configuration, for Llama (top-p=0.9), Gemma (top-k=64, top-p=0.95). When not specified by the model, we use commonly suggested values of top-k=10, top-p=0.9, min-p=0.1, and beam width=3. We set $B_{\max} = 5$ to ensure similar total allocation budgets as beam width=3, and provide sensitivity results across different beam widths and $B_{\max}$'s in the experiments. For majority voting and best-of-$n$, we use $n = B_{\max}$ to allow for a similar number of token generations among the approaches.

**Evaluation Metric.** We evaluate the accuracy of the resulting model against the ground truth, in a zero-shot pass@1 fashion. For the programming dataset (HumanEval), we evaluate against all of the test cases, only considering the code correct if all of the test cases pass.

We use HuggingFace (Wolf et al., 2019) for all models and datasets. In the following sections, we analyze the accuracy, token efficiency, dynamic allocation, and sensitivity results.

## 4.2. Results

**EDEN achieves the best average rank across the benchmarks while using substantially fewer expansions than fixed-width beam search.**

**Quality** Table 1 summarizes accuracy across datasets and baselines, highlighting the overall generation quality of each method. EDEN achieves the best (lowest) rank overall (statistically significant overall difference, with Friedman p-value 0.012), consistently outperforming greedy decoding, other test-time scaling (best-of-$n$, majority voting), and sampling-based strategies (consistent across model families, Appendix B). On all four datasets, we see a significant improvement in performance from the proposed approach above greedy decoding, all sampling-based strategies, and best-of-$n$ and majority voting. Furthermore, when compared to beam search, we generally match or improve the performance with significantly fewer expansions required, as reported between parentheses in Table 1, motivating the claim that EDEN approximates higher width beam search. In Table 2, we provide a Big O breakdown of the different time and space requirements for the various approaches.

A Bayesian hierarchical analysis shows that EDEN has a 75% posterior probability of being the best method overall (rightmost column, Table 1). At the pairwise level, EDEN has an extremely high probability ($\geq 96\%$) of outperforming nearly all competing approaches, with the exception of beam search, against which EDEN is still favoured with a 77% posterior probability (but again, requiring significantly fewer expansions). We further visualize these outcomes in Figure 3, highlighting the same pattern: EDEN outperforms the alternative decoding strategies across the pairwise dominance matrix, has the highest probability of being the best, and exhibits the most favourable posterior skill distribution.

It is also important to note that the absolute accuracy is not what is important here, but rather the relative accuracy among the methods – as we utilize a 3B base model, larger models may achieve higher SOTA accuracy (discussed in limitations, Appendix F), however, the key is that the proposed approach consistently improves these baseline models, and the reported base results are all in line with the results for models of similar sizes.

### Token Efficiency: Approximation of higher beam widths

To get a better understanding of how the generation quality scales with token utilisation, we vary the maximum branching budget $B_{\max} \in \{3, 5, 7, 9\}$ (or beam size in the comparison) and evaluate the resulting accuracy versus total number of expansions generated. This illustrates how efficiently each decoding strategy converts sampling into high-quality generations. For this, we use HumanEval for comparison. The results are shown in Figure 4. Across the compute budgets, the proposed approach consistently

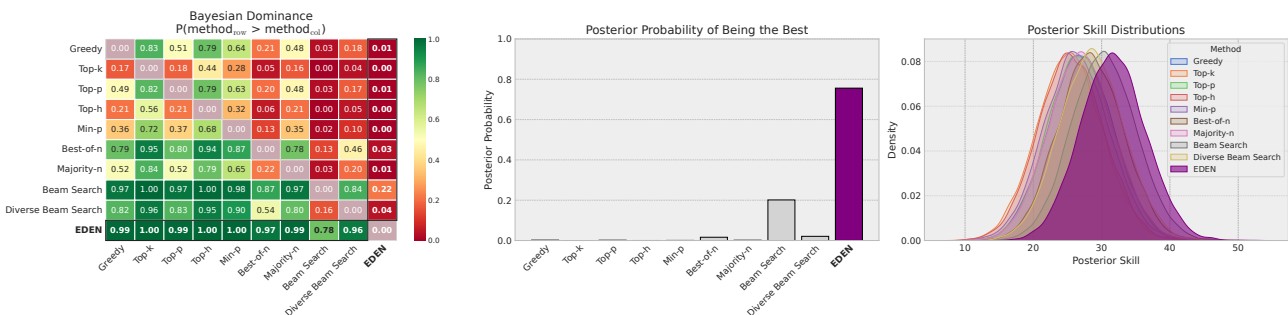

*Figure 3.* **EDEN has the strongest posterior performance across decoding methods.** Bayesian hierarchical model analysis of decoding methods. *Left:* Pairwise dominance probabilities $P(\text{method}_{\text{row}} > \text{method}_{\text{col}})$, showing that EDEN outperforms all competing approaches with high posterior probability. *Middle:* Posterior probability of each method being the best overall, with EDEN achieving the highest value by a substantial margin. *Right:* Posterior skill distributions for all methods, with EDEN achieving the highest.

*Table 1.* **EDEN achieves the best overall rank**. Performance comparison presenting the accuracy $\pm$ 95% bootstrapped half confidence interval. For the search-based approaches, the total number of branches explored is shown in brackets. The average Friedman rank is shown in the *rank* column (statistically significant difference, $p = 0.012$). The rightmost column shows the probability from a Bayesian Hierarchical Model that EDEN outperforms each competitor.

| Method | GSM8K ↑ | MATH500 ↑ | HumanEval ↑ | SciBench ↑ | *Rank* ↓ | P(EDEN > competitor) |
|---|---|---|---|---|---|---|
| Greedy | 73.5% ± 2.4% | 27.4% ± 3.8% | 27.0% ± 7.1% | 4.9% ± 1.7% | 6.12 | 0.99 |
| Top-$k$ Sampling | 70.7% ± 2.5% | 23.0% ± 3.7% | 27.6% ± 6.7% | 4.9% ± 1.6% | 7.00 | 1.00 |
| Top-$p$ Sampling | 73.5% ± 2.4% | 27.4% ± 4.0% | 27.0% ± 6.7% | 4.8% ± 1.6% | 6.62 | 0.99 |
| Top-$H$ Sampling | 69.7% ± 2.5% | 26.0% ± 3.8% | 27.0% ± 6.7% | 4.5% ± 1.5% | 8.12 | 1.00 |
| Min-$p$ Sampling | 72.3% ± 2.5% | 28.0% ± 3.8% | 25.8% ± 6.7% | 4.3% ± 1.5% | 8.00 | 1.00 |
| Best-of-$n$ (5) | 78.2% ± 2.3% | 28.2% ± 3.8% | 27.0% ± 7.1% | 5.2% ± 1.7% | 4.62 | 0.96 |
| Majority-$n$ (5) | 78.7% ± 2.2% | 30.8% ± 4.1% | 19.6% ± 5.8% | 4.1% ± 1.4% | 6.25 | 0.99 |
| Diverse Beam Search (3) | 77.5% ± 2.2% (1200) | 30.0% ± 4.0% (1200) | 26.4% ± 6.7% (1200) | 5.4% ± 1.7% (1200) | 4.75 | 0.96 |
| Beam Search (3) | **81.5% ± 2.2%** (1200) | 29.2% ± 4.0% (1200) | 28.8% ± 7.1% (1200) | 6.9% ± 1.8% (1200) | 2.25 | 0.78 |
| **EDEN** (Ours) | 80.5% ± 2.2% (598) | **32.8% ± 4.0%** (840) | **31.3% ± 7.4%** (965) | **7.4% ± 2.0%** (1018) | **1.25** | $\text{P}_{\text{best}}$=0.75 |

*Table 2.* Complexity and GPU Suitability of the methods. $T$ = generation length, $C$ = per-token forward cost, $b$ = beam width, $n$ = number of samples, and $B_{\max}$ = EDEN's maximum branching factor. $K$ denotes the per-token memory cost.

| Method | Time (Worst) | Time (Avg.) | Memory | GPU Suit. |
|---|---|---|---|---|
| Greedy | $O(TC)$ | $O(TC)$ | $O(TK)$ | Excellent |
| Top-$k$/Top-$p$ | $O(TC)$ | $O(TC)$ | $O(TK)$ | Excellent |
| Best-of-$n$ | $O(nTC)$ | $O(nTC)$ | $O(TK) \ldots O(nTK)$ | High |
| Majority Vote | $O(nTC)$ | $O(nTC)$ | $O(TK) \ldots O(nTK)$ Sequential ... Parallel | High |
| Beam Search ($b$) | $O(bTC)$ | $O(bTC)$ | $O(bTK)$ | Good |
| **EDEN** | $O(B_{\max}TC)$ | $O\left(\left(\sum_t^T B_t\right)C\right)$ | $O(B_{\max}TK)$ | Fair |

achieves improved performance with fewer expansions than beam search, essentially approximating a higher branching factor search with far fewer model generations required, providing empirical support to the theoretical claims made in Section 3.1.1. EDEN reduces *average-case* expansions compared to beam search while retaining the same worst-case asymptotic complexity (Table 2). However, we also note that due to the heterogeneous branching rates, this can make parallelization more difficult on a GPU. In terms of wall clock time, in Appendix C.1, we highlight the small over-

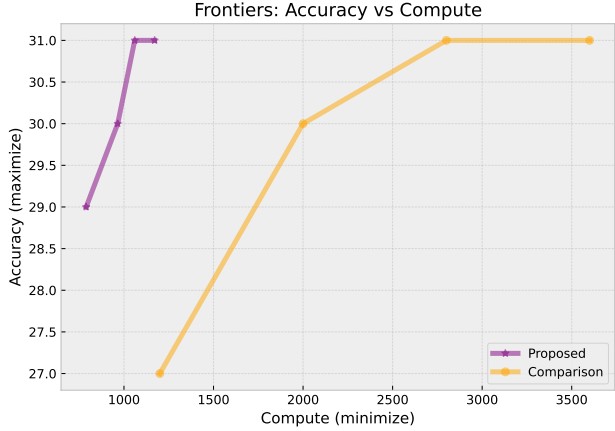

*Figure 4.* **EDEN shifts the accuracy–expansion frontier upward and leftward:** it achieves higher HumanEval accuracy with fewer expansions than fixed-width beam search.

head entropy calculation has in comparison to the forward passes saved.

**Dynamic allocation** A key benefit of the proposed approach

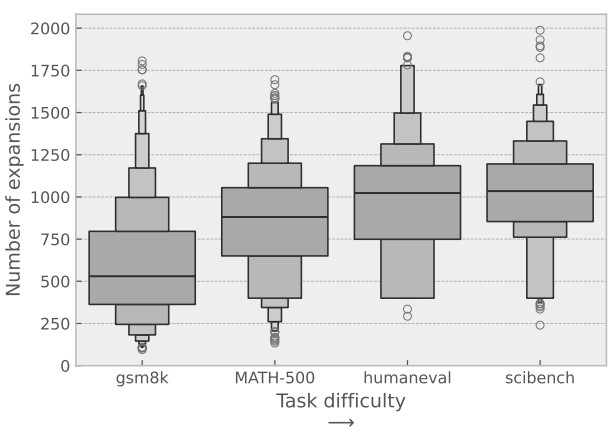

*Figure 5.* **EDEN automatically allocates more expansions to harder tasks**: without observing task labels, rewards, or ground-truth difficulty during decoding, EDEN dynamically expands more on more difficult (lower-accuracy) benchmarks.

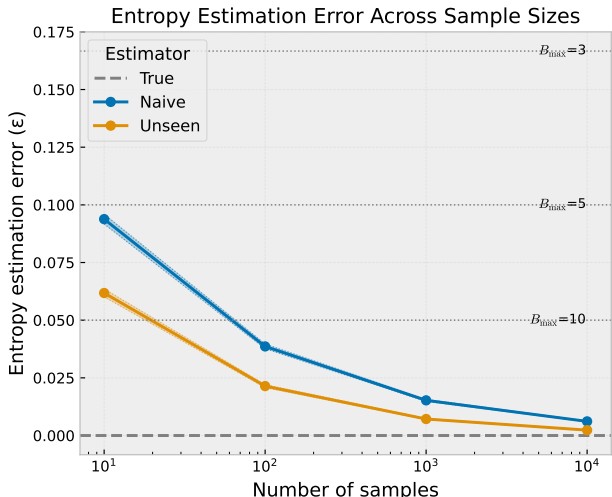

*Figure 6.* **Entropy estimation is accurate enough for EDEN branching with few samples**. RMSE of first-token entropy estimation across the datasets; dashed lines show permissible error thresholds for different $B_{max}$

is the automatic dynamic allocation of computation based on the task complexity. We can observe this dynamic allocation from the mean expansions in parentheses in Table 1, and the distributions visualised in Figure 5 (where task complexity is determined based on the resulting baseline *accuracy* from Table 1). The simplest (i.e. highest accuracy) task (GSM8K) receives the lowest number of expansions, and with the increasing task difficulty, we see that the number of expansions increases automatically up to the most difficult task on SciBench. Again, it is worth emphasizing that during the decoding process, a reward model/accuracy/difficulty measure is *not* used; these additional expansions are purely driven by the entropy of the outputs. Unlike most search methods, which expend significant compute regardless of task complexity, here, compute (in the form of expansions) is only allocated at decoding time as needed, helping to alleviate one of the main criticisms of search-based decoding, unnecessary expansions.

### 4.3. Entropy estimation under closed-API access

We next analyze entropy estimation in closed source (black box) settings. For open-source models, entropy $H$ is computed directly from $P(\mathcal{V}|x)$. When we do not have access to the full output distribution (e.g., with closed source models or limited API access), we instead approximate $\hat{H}(\mathbf{y}_{1:t}) \approx \bar{H}(\mathbf{y}_{1:t})$. A central question is: *How many samples are required to estimate the entropy of an unknown $P(\mathcal{V}|x)$ within some additive error $\epsilon$?* While a naive plugin estimator requires linear (in vocab size) $\Theta(|\mathcal{V}|/\epsilon)$ samples, optimal estimators achieve a sublinear rate $\Theta(|\mathcal{V}|/(\epsilon \log |\mathcal{V}|))$ (Valiant & Valiant, 2013; Wu & Yang, 2016). Since EDEN only uses entropy for branching decisions, it tolerates error up to $\epsilon < 0.5/B_{max}$ as rounding $M \cdot H$ to the nearest integer yields the same result whenever $|H - \hat{H}| < \frac{0.5}{B_{max}}$. With typ-

ical $B_{\max} \lesssim 10$, even small sample sizes provide sufficient estimation accuracy (see Figure 6).

Furthermore, many decoding schemes (e.g., top-$k$) restrict the effective support for the output distribution, reducing sample complexity to $\Theta(k/(\epsilon \log k))$, where $k \ll |\mathcal{V}|$. Thus, when integrating EDEN with such strategies, entropy estimation becomes even more tractable. In fact, many closed source APIs, including OpenAI, expose the top-$k$ log probs (for $k \leq 20$), allowing exact entropy computation for that subset. In such cases, EDEN can be trivially integrated into closed source LLMs under top-$k$ decoding, by searching only over the top-$k$ logits. This truncated entropy over top-$k$ is a biased underestimate of full-vocabulary entropy because tail mass is omitted; however, EDEN only requires a stable monotone proxy for branching, preserving enough of the signal for effective allocation, as demonstrated below.

**Simulated closed-API setting** We apply EDEN on top of the OpenAI API in combination with top-$k$ decoding. Specifically, we compare three settings: greedy responses from the API, top-$k$ decoding over the responses, and EDEN applied to the top-$k$ log probs. Due to high access costs and licensing constraints of closed source models, we simulate a black-box, closed source setting using Llama-3.2-3B-Instruct (discussed in limitations, Appendix F). Although Llama itself is open source, here it's treated as closed and accessed only via the OpenAI API, exposing only partial output information, mimicking the constraints of commercial black-box APIs.

By default, the OpenAI API (for GPT) provides the top $\leq 20$ log probs, so we consider $k \leq 20$. Figure 7 illustrates

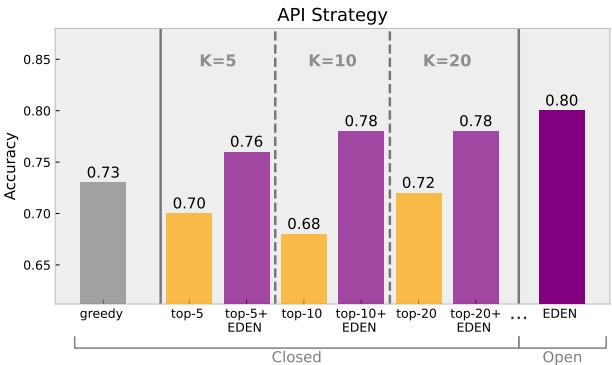

*Figure 7.* **EDEN is compatible with closed APIs** under top-$k$ decoding. EDEN improves over both direct API decoding (greedy) and standard top-$k$ decoding. Performance under restricted top-$k$ access approaches the full-logit EDEN setting as $k$ increases. The rightmost bar shows EDEN with full logit access for reference.

EDEN's integration in this setting: compared to greedy and top-$k$ decoding, EDEN applied to top-$k$ logits yields improved accuracy, demonstrating both the effectiveness of the approach and compatibility with closed source models and alternate sampling strategies. Importantly, this holds across $k$, with the added benefit of consistently improving over both top-$k$ and greedy due to EDENs adaptive nature, while in this setting, top-$k$ on its own struggles to outperform greedy decoding. The rightmost bar in Figure 7 shows the open access case, assuming full model logits are known and is used purely for comparison purposes, to show that as $k$ increases, we begin to approximate this open case.

### 4.4. Summary of Empirical Findings

Across four diverse benchmarks (covering mathematical reasoning, code generation, and scientific QA), EDEN consistently outperforms widely used decoding baselines (Table 1). Compared to greedy decoding and standard or entropy-aware sampling (top-k, top-p, min-p, Top-$H$), EDEN yields between $+2$–$11\%$ absolute accuracy improvements, while matching or exceeding beam search accuracy with significantly fewer expansions. These results are relatively consistent across model families (Appendix B). Efficiency experiments confirm that EDEN achieves a better expansion–quality frontier than beam search (Figure 4), approximating high-width search with substantially fewer expansions required, with the additional benefit of dynamically adjusting such expansions based on task difficulty (Figure 5). Sensitivity studies show that entropy estimates remain robust under limited sampling (Figure 6), paraphrasing (Figure 9), and miscalibration (Appendix D.2), with branching decisions stable within generous error bounds. Experimentally, we validated the central claim: adaptively allocating compute in proportion to entropy yields better accuracy–token trade-offs than existing decoding strategies,

and allows automatic information-based computation allocation.

## 5. Conclusion

We introduced *Entropy-informed DEcodiNg* (EDEN), a principled search-based decoding framework for adaptive information-driven branching. Our theoretical analysis establishes new entropy-based guarantees: a monotone allocation theorem showing that entropy-aware branching improves over fixed-width strategies, together with explicit regret rates and sample-complexity guarantees for reliable selection under limited rollouts. These results provide a provably justified alternative to the ad hoc thresholding and heuristic rules that dominate existing decoding approaches. Empirically, EDEN consistently outperforms standard sampling and search baselines across mathematical reasoning, code generation, and scientific QA. The proposed approach matches or exceeds the accuracy of beam search while using significantly fewer expansions, and maintains robust branching decisions even under noisy entropy estimates. Together, these findings confirm that allocating compute adaptively in proportion to a model's output entropy yields superior accuracy–expansion trade-offs compared to static approaches. EDEN opens new directions for information-aware decoding in large-scale language models. Future work may further explore integrating entropy-informed branching with custom scoring (such as diversity guided, Vijayakumar et al. (2016; 2018)) or reward functions (e.g. process reward models, Zhang et al. (2025b)), or moving beyond tokens to higher-level output abstractions such as tool calling (Yu et al., 2025) or meanings (Farquhar et al., 2024).

## Disclaimer

## Impact Statement

This paper presents work whose goal is to advance the field of Machine Learning. There are many potential societal consequences of our work, none which we feel must be specifically highlighted here.

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

## A. Notation

| Symbol | Meaning |
|--------|---------|
| $\mathcal{V}$ | Vocabulary set, with $|\mathcal{V}|$ tokens in total |
| $y$ | A token from the vocabulary $V$ |
| $\mathbf{y}_{1:t}$ | Partial output sequence of length $t$ |
| $x$ | Input/context provided to the model |
| $P(y_{t+1} \mid x, \mathbf{y}_{1:t})$ | Model distribution over the next token given context $x, \mathbf{y}_{1:t}$ |
| $\mathbf{p} = (p_1, \ldots, p_V)$ | Sorted next-token probabilities, $p_1 \geq p_2 \geq \cdots \geq p_V$ |
| $i, j$ | Indices referring to candidate tokens in $\mathcal{V}$ |
| $T$ | Maximum generation length |
| $H_t$ | Shannon entropy of the next-token distribution at step $t$: $H_t = -\sum_i p_i \log p_i$ |
| $\bar{H}_t$ | Normalised Shannon entropy ($\frac{H_t}{\log |\mathcal{V}|}$) |
| $\hat{H}_t$ | Estimated entropy from samples (used in closed source settings) |
| $\text{PP}_t = \exp(H_t)$ | Perplexity at step $t$ |
| $s(\mathbf{y}_{1:t})$ | Cumulative log-probability of a sequence: $\sum_{i=1}^{t} \log P(y_i \mid x, \mathbf{y}_{1:i-1})$ |
| $\text{Score}_\alpha(\mathbf{y}_{1:t})$ | Length-normalized score $s(\mathbf{y}_{1:t})/t^\alpha$, where $\alpha$ is a penalty exponent |
| $\alpha$ | Length penalty exponent (e.g. $\alpha = 1$ normalizes by sequence length) |
| $S^*$ | Best score among sequences explored so far (global lower bound for pruning) |
| $V_t(i)$ | Value of choosing token $i$ at step $t$: $\log P(i \mid x_t) + \text{OPT}_{t+1}(i)$ |
| $\text{OPT}_{t+1}(i)$ | Optimal continuation value if token $i$ is chosen at step $t$ |
| $i^*$ | Index of the optimal next token: $i^* = \arg\max_i V_t(i)$ |
| $\Delta_t(i)$ | Gap between the best token and candidate $i$: $V_t(i^*) - V_t(i) \geq 0$ |
| $\Delta_t^{\text{eff}}$ | Effective gap between best token and alternatives |
| $B_t$ | Adaptive branching factor at step $t$, derived from entropy $H_t$ |
| $B_{\max}$ | Maximum allowed branching factor (maximum beam width) |
| $f(H, B_{\max})$ | Scaling function converting entropy $H$ into a branching factor |
| $m_t$ | Effective compute/sampling budget allocated at step $t$ |
| $M$ | Total compute/sampling budget across all steps |
| $\overline{S}(\mathbf{y}_{1:t})$ | Upper bound on normalized score of a partial sequence |
| $\underline{S}(\mathbf{y}_{1:t})$ | Lower bound on normalized score of a partial sequence |
| $S(\mathbf{y}_{1:t})$ | Normalized score of a complete sequence (equals upper and lower bounds at EOS) |
| $\Lambda$ | Lipschitz constant controlling sensitivity of continuation values |
| $d(\cdot, \cdot)$ | Distance metric between distributions (e.g. total variation) |
| $\sigma^2$ | Variance proxy in sub-Gaussian noise model |
| $\delta_t$ | Tolerated probability of error at step $t$ |
| $\varepsilon$ | Mass outside the $\varepsilon$-typical set in Lemma 3.1 |
| $\epsilon$ | Additive error tolerance for entropy estimation |
| $c, C$ | Positive constants in concentration bounds |

*Table 3.* Summary of notation used throughout the paper and appendices.

We provide a summary of key notation used throughout the paper in Table 3.

## B. Additional Results

### B.1. Additional Models

To confirm the robustness of the results, we run on additional language models of varying sizes and from different families including Google Gemma (gemma-3-1b-it), and IBM Granite (granite-3.3-2b-instruct). This gives results on models of size $1B$, $2B$, and $3B$. The results are presented in Table 4, where EDEN again achieves the best rank (highest accuracy) across the decoding strategies on both Granite and Gemma, strengthening the claims in Section 4.

#### B.1.1. LARGER MODELS

As a proof-of-concept, we additionally run a subset of the data (100 examples from gsm8k) on Mistral (Mistral-7B-Instruct-v0.3), a 7B parameter model in Table 5. For computational reasons, we apply 8 bit quantization. Note that for this model, we applied a longer maximum token length of $T = 1000$, as we found the outputs were generally longer containing more

*Table 4.* Additional accuracy results (and resulting ranks, lower the better) across varying model families.

| | Method | GSM8K | MATH500 | HumanEval | SciBench | Rank |
|---|---|---|---|---|---|---|
| Gemma | Greedy | 32% | 19% | 18% | 3% | 4.25 |
| | Top-$k$ Sampling | 32% | 20% | **20%** | 3% | 2.63 |
| | Top-$p$ Sampling | 32% | 19% | 18% | 3% | 4.25 |
| | Min-$p$ Sampling | 31% | 19% | 18% | **4%** | 4.25 |
| | Beam Search (3) | **35%** | 20% | 19% | 3% | 2.33 |
| | EDEN (Ours) | **35%** | **21%** | 19% | 3% | **2.00** |
| Granite | Greedy | 64% | 23% | 18% | 3% | 4.25 |
| | Top-$k$ Sampling | 59% | 18% | **20%** | 2% | 4.75 |
| | Top-$p$ Sampling | 64% | 22% | 18% | 3% | 4.50 |
| | Min-$p$ Sampling | 65% | 25% | 18% | 3% | 3.62 |
| | Beam Search (3) | 69% | 27% | 19% | 3% | 2.50 |
| | EDEN (Ours) | **73%** | **31%** | 19% | **4%** | **1.37** |

*Table 5.* Additional results on a 7B Mistral model (Mistral-7B-Instruct-v0.3)

| | EDEN | Beam | Top-k | Top-p | Min-p | Greedy |
|---|---|---|---|---|---|---|
| **GSM8K** | 47% | 46% | 39% | 37% | 42% | 37% |

reasoning than the smaller models, so this gave sufficient space to generate final answers. Again, we can see the proposed approach performs strongly, achieving the highest accuracy.

## B.2. Alternative Scorers: Reward Models

*Table 6.* Augmenting with external reward model on a subset (50) of MATH-500 examples

| | $\lambda = 0$ | $\lambda = 0.05$ | $\lambda = 0.1$ |
|---|---|---|---|
| **MATH-500** | 32% | 36% | 36% |

We investigate using a custom scorer (see Appendix E.4) instead of just the log likelihood. Specifically, we use the Qwen/Qwen2.5-Math-PRM-7B (Zhang et al., 2025b) process reward model (PRM) to assign bonus per step rewards $0 \leq \rho_t \leq 1$. We perturb the log-likelihood score $F_{\mathrm{LL}}$ to include a bonus term:

$$F = F_{\mathrm{LL}} + \lambda R, \qquad R(y_{1:T}) = \sum_{t=1}^{T} \rho_t(y_t, x, y_{1:t-1}).$$

and vary the bonus strength parameter to show the impact of guiding the base model (Llama-3.2-3B-Instruct) through an external reward model $\rho$. Specifically, we look at $\lambda \in \{0, 0.05, 0.1\}$. The results are displayed in Table 6, where we can see an improvement in accuracy (+4%) when including a bonus term, controlling the stepwise process towards the desirable outcome as governed by the process reward model.

## C. Additional Algorithm Details

### C.1. Resources

We present an overview of the big-O costs and GPU suitability in Table 2. EDEN reduces *average-case* expansions compared to beam search while retaining the same worst-case asymptotic complexity as standard beam search, with a small constant-factor $O(\mathcal{V})$ overhead from calculating entropy (compared to the quadratic forward pass in $T$), shown in Figure 8. EDEN's main savings come from reducing the average active beam size $B_t \leq B_{\max}$, reducing the amount of forward passes when compared to beam search. However, we note that this is a reduction in *token usage*, and not necessarily wall clock time.

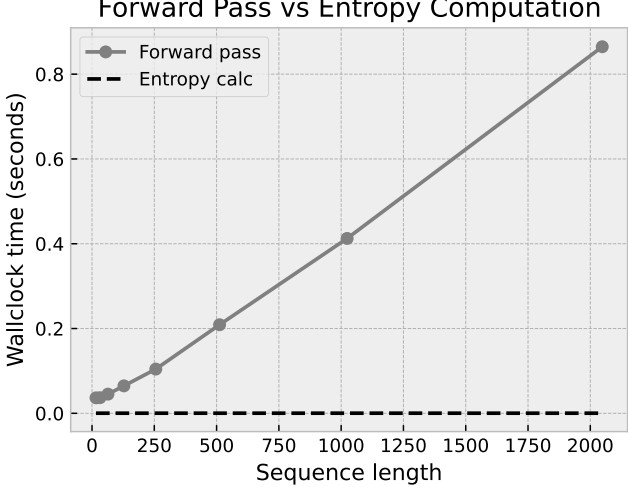

*Figure 8.* Overhead of computing entropy versus performing a forward pass

### C.2. Pruning

For efficient and admissible pruning of low probability sequences, EDEN utilizes a branch-and-bound style search algorithm (Rush et al., 2013). To prune the search space efficiently, we compute admissible upper and lower bounds on the normalized score that a partial sequence can achieve (as presented in Algorithm 1), and if an upper bound does not exceed the best lower bound seen, we eliminate the path to focus on the most promising regions.

**Upper bound.** The best-case score assumes all future tokens for a partial sequence $\mathbf{y}_{1:t}$ are maximally probable ($P = 1$). Then, the upper bound on the normalized score (at $t$) is:

$$\overline{S}(\mathbf{y}_{1:t}) = \frac{s(\mathbf{y}_{1:t}) + (T - t) \cdot \log(1)}{T^{\alpha}} = \frac{s(\mathbf{y}_{1:t})}{T^{\alpha}}.$$

**Lower bound.** The worst-case score assumes the remaining tokens in the sequence have the lowest probability (for the top-token), which is based on the size of the vocabulary ($P = \frac{1}{V}$):

$$\underline{S}(\mathbf{y}_{1:t}) = \frac{s(\mathbf{y}_{1:t}) + (T - t) \cdot \log\left(\frac{1}{V}\right)}{T^{\alpha}}.$$

**Completed sequences.** If a partial sequence ends with the end-of-sequence token (EOS) (or $t = T$), we treat it as complete. In this case, both the upper and lower bounds reduce to its actual normalized score:

$$\overline{S}(\mathbf{y}_{1:t}) = \underline{S}(\mathbf{y}_{1:t}) = \frac{s(\mathbf{y}_{1:t})}{t^{\alpha}}.$$

**Running lower bound.** We maintain a global best lower bound across all sequences seen so far:

$$S^* = \max_{\mathbf{y}_{1:t} \in \mathcal{S}} \underline{S}(\mathbf{y}_{1:t}),$$

where $\mathcal{S}$ is the set of all explored sequences. $S^*$ is initialized using the score of the greedy decoding sequence, giving a strong lower bound at the beginning of the search, allowing poor sequences to be eliminated early.

**Pruning rule.** A candidate sequence $\mathbf{y}_{1:t}$ is pruned if its upper bound falls below the best known lower bound:

$$\overline{S}(\mathbf{y}_{1:t}) < S^*.$$

This ensures that only candidates who could potentially outperform the current best are retained. The use of admissible bounds, where the upper bound never underestimates and the lower bound never overestimates, preserves optimality within the explored search space, ensuring we never eliminate potential best-scoring paths.

# D. Additional Robustness Discussion

## D.1. Robustness to Prompt Paraphrasing

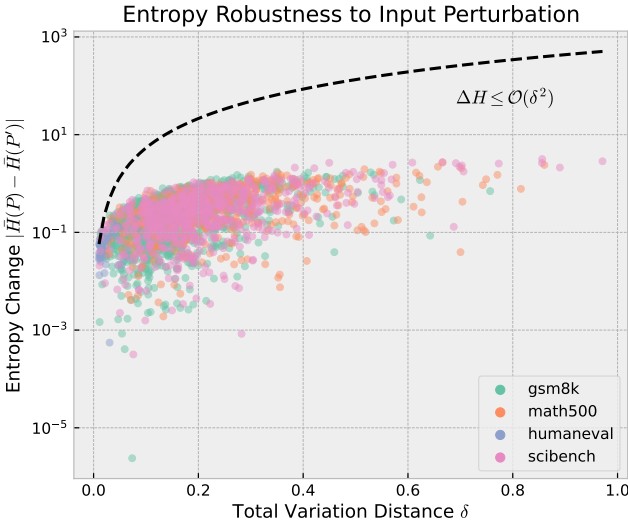

*Figure 9.* Stability of entropy under input perturbations (Appendix D.1). Each point corresponds to a randomly perturbed (in embedding space) prompt. Small total variation shifts in $P(\mathcal{V}|x)$ induce only minor changes in $H$, ensuring robust branching factors.

We test EDEN's stability through perturbing the embedding space of the prompts, measuring the variance in the resulting entropy. The entropy $H(P(\cdot \mid x))$ is a stable function of the input $x$, assuming the LLM exhibits a smooth mapping from inputs to output distributions. In particular, if $x'$ is a perturbed version of $x$ such that the total variation distance between their output distributions satisfies $\|P(\cdot \mid x) - P(\cdot \mid x')\|_1 \leq \delta$, then the change in entropy is bounded by $|H(P(\cdot \mid x)) - H(P(\cdot \mid x'))| \leq D_{\mathrm{KL}}(P(\cdot \mid x)\|P(\cdot \mid x')) + D_{\mathrm{KL}}(P(\cdot \mid x')\|P(\cdot \mid x)) \leq \mathcal{O}(\delta^2)$, by Pinsker's inequality and the continuity of entropy. This ensures that minor input variations (such as paraphrasing, reordering, or synonym substitutions) lead to only small differences in entropy, and therefore do not substantially affect branching decisions, making the entropy-based splitting criteria used in our method stable and reliable. The robustness to variations is demonstrated in Figure 9, showing the empirical relationship between total variation distance and entropy change for perturbed inputs to the LLM (Llama-3.2-3B-Instruct) across all datasets. As predicted by theory, entropy differences remain small when the total variation distance between output distributions is small, and grow subquadratically, consistent with the $\mathcal{O}(\delta^2)$ bound from Pinsker's inequality. This illustrates the robustness of entropy under small input perturbations, supporting its use as a stable criterion for branching decisions.

## D.2. Robustness to Miscalibration

It is well documented that LLM logits are often miscalibrated, i.e. their predicted probabilities do not align with true correctness likelihoods (Lovering et al., 2025). However, EDEN does not require perfectly calibrated probabilities: its purpose is not to recover real-world frequencies but to guide exploration according to the model's own internal distribution.

EDEN relies on the *relative shape* of the predictive distribution $P(\cdot \mid x, \cdot)$, rather than on absolute calibration. A low-entropy (peaked) distribution signals that the model strongly prefers a small set of tokens, while a high-entropy (flat) distribution signals uncertainty. These structural cues remain valid regardless of whether probabilities are systematically under- or over-confident. Branching decisions depend on discrete thresholds of entropy (e.g. $B_t = \lfloor B_{\max} \cdot H_t \rfloor$). Small shifts in calibration that perturb entropy by less than $1/B_{\max}$, do not change the branching factor. Thus, allocation decisions are relatively stable even under moderate miscalibration.

**Stability under temperature scaling.** Temperature scaling, a common form of calibration adjustment (Guo et al., 2017), applies a monotonic transformation to the logits, predictably altering entropy, since when applying temperature scaling, the

distribution becomes:

$$P_i^{(\mathcal{T})} = \frac{P_i^{1/\mathcal{T}}}{\sum_j P_j^{1/\mathcal{T}}},$$

i.e., $H$ strictly increases with temperature. As $\mathcal{T} \to 0$, $H \to 0$ (deterministic), and as $\mathcal{T} \to \infty$, $H \to \log |\mathcal{V}|$ (uniform uncertainty). The derivative $\frac{dH}{d\mathcal{T}}$ tells us exactly how entropy and thus the branching factor $B_t$ scales with temperature in a controlled manner: increasing $\mathcal{T}$ increases entropy (and branching), while lowering $\mathcal{T}$ tightens focus in a quantified and stable way. Since our method tolerates entropy errors up to $\frac{0.5}{B\max}$ (and $f$ is monotone with $H$), these variations are typically well within safe bounds, ensuring that branching decisions remain stable under common forms of miscalibration or numerical noise.

**Practical implications.**   Our objective is not to align model probabilities with external truth, but to exploit the model's internal confidence structure to adaptively allocate computation. Entropy is a reliable proxy for this confidence: it highlights points of high ambiguity where additional branching is useful, and remains stable under the typical forms of calibration shift encountered in practice. This robustness ensures that EDEN maintains relatively consistent behavior across models and decoding settings, without requiring any explicit calibration correction.

## E. Additional Theory

### E.1. Sample complexity

**Sub-Gaussian noise.** An estimator $\hat{V}$ of a random variable $V$ is $\sigma^2$-sub-Gaussian if

$$\Pr(|\hat{V} - \mathbb{E}[V]| > \epsilon) \le 2 \exp\left(-\frac{m\epsilon^2}{2\sigma^2}\right)$$

when computed from $m$ independent samples. Constants $c, C > 0$ in the main text depend only on this variance proxy.

**Proposition E.1** (Sample complexity for correct selection). *Suppose value estimates are sub-Gaussian with variance proxy $\sigma^2/m$ when using $m$ samples. To guarantee probability $\le \delta$ of selecting a non-optimal element among $s$ candidates with effective gap $\Delta_{\text{eff}}$, it suffices that*

$$m \ge \frac{C}{\Delta_{\text{eff}}^2}\left(\log s + \log \frac{1}{\delta}\right),$$

*where $C > 0$ depends only on the sub-Gaussian constant.*

*Proof.* For each $i \ne i^*$, concentration gives $\Pr(\hat{V}(i) \ge \hat{V}(i^*)) \le e^{-cm\Delta_{\text{eff}}^2}$. By a union bound, the error probability is at most $se^{-cm\Delta_{\text{eff}}^2}$. Requiring this to be $\le \delta$ yields the claimed condition with $C = 1/c$. $\qquad\square$

### E.2. Adaptive outperforms fixed for a fixed computation budget

Assume that with $m_t$ samples at step $t$, the per-step error proxy satisfies:

$$\Pr(\text{mistake at } t) \ \le \ A_t\, e^{-\kappa_t m_t},$$

with $A_t \asymp \text{PP}_t$ and $\kappa_t \asymp c(\Delta_t^{\text{eff}})^2$. From Proposition 3.3, $\text{PP}_t$ increases monotonically with entropy $H_t$ and $\Delta_t^{\text{eff}}$ decreases monotonically with $H_t$, so $A_t$ is increasing and $\kappa_t$ is decreasing in $H_t$.

With total budget $M$, the convex program

$$\min_{m_t \ge 0,\, \sum m_t = M} \ \sum_{t=1}^{T} A_t\, e^{-\kappa_t m_t}. \tag{$\star$}$$

has a unique minimizer $m^*$, as the objective is a sum of strictly convex functions of $m_t$. The allocation $m^*$ is non-decreasing in $H_t$, and any monotone function $m_t \propto \phi(H_t)$ strictly improves over equal allocation $m_t = M/T$ whenever the $H_t$ are not all equal.

*Proof.* Each term $A_t e^{-\kappa_t m_t}$ is convex in $m_t$. The Lagrangian stationarity gives $-A_t \kappa_t e^{-\kappa_t m_t} + \lambda = 0$, so

$$m_t^* = \frac{1}{\kappa_t} \log\left(\frac{A_t \kappa_t}{\lambda}\right).$$

where $\lambda > 0$ is chosen so that $\sum_t m_t^\star = M$. As $A_t$ increases and $\kappa_t$ decreases with $H_t$, the RHS increases in $H_t$. Thus $m^*$ is monotone. Equal allocation is optimal only if all $A_t \kappa_t$ are constant across $t$; otherwise $m^*$ strictly outperforms it. $\quad\square$

**Theorem E.2** (Entropy-adaptive dominates fixed allocation). *If $H_t$ are not all equal, then the entropy-adaptive allocation*

$$m_t^{\mathrm{EA}} \;\propto\; \kappa_t(H_t)^{-1} \log\big(A_t(H_t)\,\kappa_t(H_t)\big)$$

*(rescaled to $\sum m_t^{\mathrm{EA}} = M$) coincides with the unique minimizer $m^\star$ of equation $\star$ (by strict convexity) and hence*

$$\underbrace{\sum_{t=1}^{T} A_t e^{-\kappa_t m_t^{\mathrm{EA}}}}_{\textit{Adaptive}} \;<\; \underbrace{\sum_{t=1}^{T} A_t e^{-\kappa_t (M/T)}}_{\textit{Fixed}}.$$

*Proof.* Since equation $\star$ is strictly convex, its minimizer $m^\star$ is unique. If $H_t$ are not all equal (which is assumed to be the case in any well-trained LLM, and empirically verified for all tasks here), then $A_t \kappa_t$ is not constant across $t$ (by Proposition 3.3), so the fixed allocation $m_t = M/T$ does not satisfy the KKT optimality conditions. Therefore $m^\star \neq M/T$ and strict convexity implies

$$\underbrace{\sum_{t=1}^{T} A_t e^{-\kappa_t m_t^\star}}_{\textit{Optimal}} \;<\; \underbrace{\sum_{t=1}^{T} A_t e^{-\kappa_t (M/T)}}_{\textit{Fixed}}.$$

Since $m_t^{\mathrm{EA}}$ coincides with $m^\star$, the result follows. $\quad\square$

**Corollary E.3** (Robust advantage of monotone policies). *For large enough $M$, any policy $m_t^\phi \propto \phi(H_t)$ with $\phi$ monotone and bounded distortion of $A_t, \kappa_t$ achieves a constant-factor improvement over the fixed policy whenever $\mathrm{Var}(H_t) > 0$.*

*Proof.* If $\phi$ distorts $A_t$ and $\kappa_t$ by at most fixed multiplicative constants, the KKT solution is perturbed only by bounded factors. Since $H_t$ are not all equal, the resulting allocation deviates from $M/T$ on a positive-measure subset of steps, reducing the objective by a multiplicative constant. This constant factor persists as $M \to \infty$ whenever $\mathrm{Var}(H_t) > 0$. $\quad\square$

### E.2.1. EXPLICIT CUMULATIVE REGRET UNDER A UNIFORM GAP CONDITION

We now make explicit the cumulative regret rate implied by the per-step error control used in Theorem 3.4 and Appendix E.1.

**Proposition E.4** (Cumulative regret under a uniform gap condition). *Suppose that, at each step $t = 1, \ldots, T$, the score estimates used for selection are sub-Gaussian, as in Proposition E.1. Let $i_t^*$ denote the optimal action at step $t$, let $I_t$ be the selected action, and define the instantaneous regret*

$$\Delta_t(i) = r_t(i_t^*) - r_t(i).$$

*Assume that there exist constants $\Delta_{\min} > 0$, $G > 0$, and $P_{\max} > 0$ such that*

$$\Delta_t^{\mathrm{eff}} \geq \Delta_{\min}, \qquad 0 \leq \Delta_t(i) \leq G, \qquad \mathrm{PP}_t \leq P_{\max}$$

*for all $t$ and all candidate actions $i$. Then, for some constant $c > 0$, the expected cumulative regret*

$$R_T = \sum_{t=1}^{T} \Delta_t(I_t)$$

*satisfies*

$$\mathbb{E}[R_T] \leq G P_{\max} \sum_{t=1}^{T} \exp\big(-c m_t \Delta_{\min}^2\big).$$

*Proof.* By the sub-Gaussian selection bound from Proposition E.1, the probability of selecting a suboptimal action at step $t$ is bounded by

$$\Pr(I_t \neq i_t^*) \leq \mathrm{PP}_t \exp\big(-cm_t(\Delta_t^{\mathrm{eff}})^2\big).$$

Using $\mathrm{PP}_t \leq P_{\max}$ and $\Delta_t^{\mathrm{eff}} \geq \Delta_{\min}$ gives

$$\Pr(I_t \neq i_t^*) \leq P_{\max} \exp\big(-cm_t\Delta_{\min}^2\big).$$

Since the instantaneous regret is bounded by $G$,

$$\mathbb{E}[\Delta_t(I_t)] = \sum_{i \neq i_t^*} \Delta_t(i) \Pr(I_t = i) \leq G \Pr(I_t \neq i_t^*).$$

Therefore,

$$\mathbb{E}[\Delta_t(I_t)] \leq GP_{\max} \exp\big(-cm_t\Delta_{\min}^2\big).$$

Summing over $t = 1, \dots, T$ yields the claim. $\qquad\square$

**Corollary E.5** (Regret rates under common budget scalings). *Under the assumptions of Proposition E.4, the following rates hold.*

*Constant budget. If $m_t \equiv m$, then*

$$\mathbb{E}[R_T] \leq TGP_{\max} \exp\big(-cm\Delta_{\min}^2\big).$$

*Logarithmic budget. If*

$$m_t \geq \frac{\alpha}{\Delta_{\min}^2} \log T$$

*for all $t$, then*

$$\mathbb{E}[R_T] = O\big(T^{1-c\alpha}\big).$$

*In particular, if $c\alpha > 1$, the regret is sublinear.*

*Critical scaling. If*

$$m_t \geq \frac{1+\epsilon}{c\Delta_{\min}^2} \log T,$$

*then*

$$\mathbb{E}[R_T] = O(1),$$

*i.e., the regret is bounded.*

*Proof.* The *constant-budget* claim follows by substituting $m_t = m$ into Proposition E.4.

For the *logarithmic-budget* claim,

$$\exp\big(-cm_t\Delta_{\min}^2\big) \leq \exp(-c\alpha \log T) = T^{-c\alpha}.$$

Summing over $T$ steps gives

$$\mathbb{E}[R_T] \leq GP_{\max}T^{1-c\alpha}.$$

The *critical-scaling* claim follows by taking $\alpha = (1+\epsilon)/c$, which gives

$$\mathbb{E}[R_T] \leq GP_{\max}T^{-\epsilon},$$

and hence, in particular, $\mathbb{E}[R_T] = O(1)$. $\qquad\square$

Thus, under a standard uniform-gap condition, the entropy-adaptive allocation guarantee can be translated into an explicit cumulative regret rate.

### E.3. Justifying the distributional Lipschitz continuity assumption

Our proposition relies on a regularity condition: small perturbations of the model's output distribution should not induce arbitrarily large jumps in future (continuation) values. This is in the spirit of the *simulation lemma* from reinforcement learning, and formalizes the smoothness intuition that similar predictive distributions yield similar downstream values.

**Why this is reasonable.** In our setting, the per-step "reward" is the normalized log-likelihood, which is bounded (e.g., via temperature scaling). LLMs typically compute next-token probabilities through (relatively) smooth maps of the hidden state due to training over *expectations* of an extremely large number of future continuations, so small perturbations of the context induce only small perturbations in $P(\cdot \mid x_t, \cdot)$, making the Lipschitz continuity with respect to the predictive distribution a realistic regularity assumption.

Empirically, minor context edits or paraphrases produce only modest changes in next-token probabilities (e.g., Figure 9), making the distributional Lipschitz continuity assumption reasonable in practice.

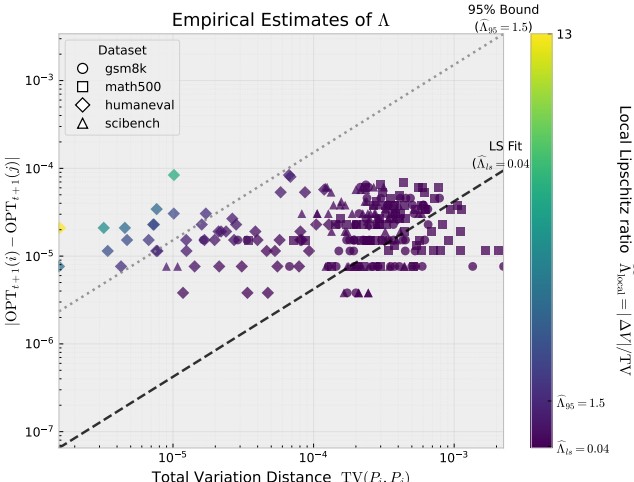

*Figure 10.* Empirical validation of the distributional Lipschitz continuity assumption. Each point corresponds to a small isotropic embedding perturbation applied to (100 equally sampled, 25 from each) prompts from GSM8K, MATH500, HumanEval, and SciBench. The plot shows the relationship between the induced total-variation shift in the next-token distribution and the absolute change in continuation log-likelihood. A least-squares fit through the origin gives $\Lambda_{\mathrm{LS}} \approx 0.04$, while the 95th-percentile local Lipschitz ratio is $\Lambda_{95} \approx 1.5$, indicating smooth dependence of continuation values on next-token distributions, helping to empirically verify the assumption.

**Empirical validation.** We empirically assess this regularity assumption by measuring the sensitivity of continuation values to small, norm-controlled perturbations of the model's next-token distribution. For prompts drawn from GSM8K, MATH500, HumanEval, and SciBench, we (i) compute input embeddings and a greedy continuation, (ii) apply small isotropic perturbations to the embeddings, (iii) measure the resulting total-variation (TV) shift in the next-token distribution, and (iv) record the absolute change in the continuation log-likelihood. Each perturbation therefore yields a pair

$$\bigl(\mathrm{TV}(P, \tilde{P}), |\Delta V|\bigr),$$

capturing the local sensitivity of continuation values to next-token distribution shifts.

As shown in Figure 10, continuation values vary *slowly* as a function of TV distance. A least-squares fit through the origin gives a global slope of $\Lambda_{\mathrm{LS}} \approx 0.04$, while the 95th-percentile local Lipschitz ratio is $\Lambda_{95} \approx 1.5$. These small slopes indicate that continuation values are highly insensitive to modest distributional perturbations.

This low-sensitivity behavior provides direct empirical support for the distributional Lipschitz continuity assumption used in our analysis. Thus the distributional Lipschitz continuity assumption is both mathematically convenient and empirically realistic for log-likelihood-based rewards. For more arbitrary or discontinuous reward functions, such an assumption may not hold. We discuss the required conditions in Appendix E.4.

### E.4. Alternative Scorers

Maximizing the log-likelihood of a resulting sequence is not always desirable. While our theoretical results above are derived for the length–penalized log-likelihood objective:

$$F_{\mathrm{LL}}(y_{1:T}) = \sum_{t=1}^{T} \frac{\log P(y_t \mid x, y_{1:t-1})}{t^\alpha},$$

we demonstrate here that the analysis extends to any sequence-level scorer that satisfies a set of structural conditions. This section specifies the required properties and clarifies when the entropy–adaptive guarantees remain valid.

**(1) Additive decomposability.** All regret and branching results rely on the scorer admitting a per-step decomposition

$$F(y_{1:T}) = \sum_{t=1}^{T} r_t(y_t, x, y_{1:t-1}), \tag{S.1}$$

which induces value functions $V_t(i) = r_t(i) + \mathrm{OPT}_{t+1}(i)$. This assumption ensures that continuation values are well-defined and that standard concentration bounds apply.

**(2) Bounded per-step rewards.** We require

$$|r_t(\cdot)| \le B \qquad \forall t, \tag{S.2}$$

which guarantees sub-Gaussian rollout estimates and underpins the safe-pruning criterion in Algorithm 1. Unbounded or heavy-tailed rewards can violate the concentration guarantees.

**(3) Distributional Lipschitz continuity.** The key regularity assumption needed for Proposition 3.3 is that future values change smoothly under small perturbations of the next-token distribution:

$$\left| \mathrm{OPT}_{t+1}(i) - \mathrm{OPT}_{t+1}(j) \right| \le \Lambda\, d\big(P(\cdot \mid x_t, i), P(\cdot \mid x_t, j)\big), \tag{S.3}$$

for some metric $d$ (e.g. total variation). Scorers with discontinuous or adversarial dependence on the predictive distribution generally violate S.3.

**(4) Entropy–dependent effective gaps.** We require the effective gap $\Delta_t^{\mathrm{eff}}$ for a step-level reward $r$ to decrease monotonically with entropy $H_t$:

$$\Delta_t^{\mathrm{eff}} = \min_{i \ne i^*} \left[ r_t(i^*) - r_t(i) - \Lambda\, d\big(P(\cdot \mid x_t, i), P(\cdot \mid x_t, i^*)\big) \right]$$

which drives the optimality of entropy-proportional allocation. For a general scorer $F$, we require only that this monotonicity holds up to a constant shift. One way to ensure this is through adding a small, bounded, Lipschitz regularizer to the log-likelihood:

$$F = F_{\mathrm{LL}} + \lambda R, \qquad R(y_{1:T}) = \sum_{t=1}^{T} \rho_t(y_t, x, y_{1:t-1}).$$

to control the decoded output towards some desirable result (encoded in $\rho$). If the regularizer satisfies

$$|\rho_t(i) - \rho_t(j)| \le C_R, \tag{S.4}$$

$$|\mathrm{OPT}_{t+1}^R(i) - \mathrm{OPT}_{t+1}^R(j)| \le \Lambda_R\, d\big(P(\cdot \mid x_t, i), P(\cdot \mid x_t, j)\big), \tag{S.5}$$

then the combined scorer inherits the entropy–gap structure:

$$\Delta_t^{\mathrm{eff}}(H_t) \ge \Delta_t^{\mathrm{eff,LL}}(H_t) - \lambda C',$$

for a horizon-dependent constant $C'$. As long as $\lambda C'$ does not dominate the LL term, the map $H_t \mapsto \Delta_t^{\mathrm{eff}}(H_t)$ remains monotone, and all entropy-adaptive allocation results continue to hold with modified constants. In practice, this allows one to control the decoded outputs towards some desirable outcome (beyond just likelihood).

The entropy-adaptive branching guarantees apply to any scorer $F$ satisfying:

1. additive, bounded per-step rewards (S.1–S.2);

2. distributional Lipschitz continuity (S.3);

3. monotone entropy–gap structure (automatically inherited for bounded log-likelihood perturbations, S.4);

4. sub-Gaussian rollout concentration (a consequence of bounded rewards).

These conditions are close to minimal: violating any one breaks at least one component of the effective-gap or regret analysis. Within this class, the entropy-adaptive allocation continues to outperform any fixed-width branching policy, as established in Section 3.1.1.

## F. Limitations

**Computation** In the paper, we look at compute based on the number of model expansions (e.g. token generations). However, this is just one view of compute. It is important to note that calculating the entropy at each step comes with its own cost. Additionally, having heterogeneous branching factors makes batching and GPU parallelism more difficult, serving as another potential limitation.

**Confidence** The tasks considered in this work generally stem around logical tasks, including math, programming and science, where generally higher confidence in the answers is preferred. However, for various other text generation tasks, such as creative writing, optimizing for sentence likelihood may not be as desirable. In this case, alternate scorers can be utilized following Appendix E.4.

**Calibration** As discussed in Appendix D.2, the approach assumes there is useful information in the learnt output distribution. For undertrained models, or those with extremely poorly calibrated outputs, the proposed approach may not provide benefits due to the lack of information in the shape of the output distribution.

**Experiments** We have favored providing a breadth of experiments across datasets, model families, and scenarios, rather than repeated runs on the same experiments. While this does still help build faith in robustness of the results, ideally, every experiment (e.g., on the comparisons) would have been repeated many times with different samples. However, again, with limited compute budget we favored getting a broader set of bootstrapped results.

**Model usage** Due to computation limitations, we have restricted our main analysis to models $\leq$ 3B parameters, with supplementary analysis up to 7B. Ideally, we would also analyse larger open-source models, but this proved computationally prohibitive even for base greedy decoding. Additionally, we only *simulated* closed source models due to licensing constraints, however, ideally we would have used a real closed source model.

