# OpenReview forum: "Entropy-informed Decoding: Adaptive Information-Driven Branching"
_ICML.cc/2026/Conference — ICML 2026 regular_

### Official Review · Reviewer_KMmU · 2026-03-09

**Soundness:** 4
**Presentation:** 4
**Significance:** 3
**Originality:** 3
**Overall Recommendation:** 5
**Confidence:** 3

**Summary:**

The authors propose EDEN, an LLM decoding framework that uses the entropy of the next-token distribution as a signal of model uncertainty during generation. When entropy is high, EDEN allocates additional compute by expanding more branches, and when entropy is low, it follows a greedier path, resulting in a variable-width beam search that focuses exploration on the most uncertain decoding steps. Branches are pruned using a branch-and-bound algorithm to retain only the most promising trajectories. The authors provide theoretical guarantees that this entropy-monotone allocation outperforms any fixed-width branching strategy under the same compute budget. Empirical results show that EDEN outperforms standard decoding baselines on math, science, and coding benchmarks, matching or exceeding beam-search accuracy while using significantly fewer expansions.

**Compliance With Llm Reviewing Policy:**

Affirmed.

**Final Justification:**

I maintain my recommendation to accept. The paper presents a clean and well-motivated framework that uses entropy as a signal for adaptive compute allocation during decoding, supported by solid theoretical development and thoughtful experiments.

The rebuttal addressed my concerns satisfactorily. The authors agreed to clarify the distinction between accuracy and efficiency gains, to make the task difficulty ordering explicit as a model-relative measure, and to discuss the scaling question with appropriate nuance. I appreciated the honest engagement with the GPU suitability point in particular. While a gap remains between algorithmic expansion savings and realized wall-clock speedups, the authors have committed to acknowledging this tension explicitly and to framing EDEN as improving the allocation of computation rather than guaranteeing proportional runtime gains. This is a reasonable and accurate framing.

Overall, the strengths of this work, including its clean theory, practical applicability without retraining or reward models, and strong experimental design, outweigh the remaining limitations. I believe this is a solid contribution to the literature on decoding and inference-time compute.

**Key Questions For Authors:**

1. Figure 5 lists "task difficulty" along the x-axis, but the ordering appears to be determined by the model's accuracy on each benchmark. How do the authors justify that HumanEval is harder than MATH-500, or that SciBench is harder than both?
2. Despite using fewer expansions than beam search, EDEN receives the lowest GPU suitability rating of all methods due to its irregular branching factors. Could the authors provide wall-clock time comparisons on the same hardware to clarify whether the reduction in expansions translates to practical speedups, or whether batching inefficiencies and the initial greedy pass erode much of that advantage?
3. The main experiments are limited to models ≤ 3B parameters, with a brief 7B proof-of-concept on a single dataset. Do the authors expect the entropy-based branching gains to hold at larger scales, where models tend to be better calibrated and distributions may be sharper overall? If entropy variance decreases with model scale, Theorem E.2 suggests the advantage over fixed-width search would shrink accordingly.

**Limitations:**

Yes.

**Strengths And Weaknesses:**

Strengths:
1. The paper is well written with a clean theoretical development. The most contentious assumptions (distributional Lipschitz continuity, monotonic entropy-gap) are carefully supported through empirical validation (Figure 10) and earlier lemmas (3.1, 3.2), making the progression to the main regret result (Theorem 3.4) easy to follow.
2. The experiments are designed well, are interesting, and support the claims from the theoretical sections.
3. EDEN is practically appealing: it requires no retraining or external reward models, works with both open-source and closed-source models, and composes with existing decoding strategies like top-k and nucleus sampling.

Weaknesses:
1. The accuracy improvements over beam search are modest and often within the reported 95% confidence intervals (e.g., 80.5% vs 81.5% on GSM8K). The primary advantage over beam search is efficiency (fewer expansions) rather than raw accuracy, but the paper's framing sometimes conflates the two, leaving the reader to disentangle these contributions.
2. The figures could use some polishing. For example, I recommend using "Beam Search" instead of "Comparison" in the legend for Figure 4, aligning the x-axis labels appropriately in Figure 3 (left), and adjusting font sizes to improve readability.
3. Figure 5 lists "task difficulty" along the x-axis, but it is ambiguous how the authors can claim HumanEval (a coding benchmark) is more difficult than MATH-500 (a math benchmark), and why SciBench (a science QA benchmark) is harder than both of those. Is this arrangement of task difficulty standard? If not, how did the authors choose the ordering?
4. Despite the fact that EDEN uses fewer branches than beam search, it has worse GPU suitability, making it the worst with respect to GPU suitability over all baselines. Therefore, the computational savings of EDEN over beam search are not clear.

Despite these weaknesses, I think this is a solid paper overall.

---

> ### Author Rebuttal · Authors · 2026-03-26
>
> We would like to thank the reviewer for their careful review, comments, and suggestions!
>
> We are pleased that the reviewer finds the paper well-written, with a clean theoretical development, and believes that the assumptions made are well-validated empirically. We appreciate the reviewer’s careful consideration of the assumptions, proofs, and overall approach.
>
> We hope that the following addresses any remaining concerns of the reviewer:
>
> * **Conflation of accuracy and efficiency**: We will rewrite parts of the discussion in Section 4.2 to make this distinction clearer
>
> * **Figures**: Thank you for the suggestion, we will update to improve the readability.
>
> * **Task difficulty**: It is true that task difficulty is a subjective measure. However, here, we used the raw accuracy (baseline, greedy decoding) from Table 1 as a proxy for the task difficulty. This removes manual judgment and relies on the model’s own performance to define the ordering in Figure 5. We will make it clearer that this is a model-relative notion of difficulty, and will clarify this explicitly to avoid implying any universal ordering.
>
> * **GPU**: EDEN has equivalent GPU suitability as beam search, as it can be executed by treating the process as a fixed B_max beam width search and masking out the unchosen branches. However, we agree that irregular branching can introduce batching inefficiencies in practice, and that wall-clock performance depends on implementation details. While EDEN reduces the number of expansions substantially (Table 1), we will clarify that the realized speedup may vary depending on implementation and system-level considerations. We opted for an implementation-agnostic complexity comparison in Table 2 to account for these implementation-specific details.
>
>
> * **Larger models**: We agree with the reviewer’s intuition that as models become better calibrated on a given benchmark, output distributions may become sharper on average, which could reduce the potential gains from adaptive branching.
> However, recent work [1] suggests that this effect is relatively modest in practice, where entropy remains non-trivial even in larger models (up to 70B), and that improvements in calibration (and the corresponding reductions in entropy) occur only very slowly with scale. This indicates that, even at larger model sizes, there remains meaningful uncertainty in next-token distributions.  Additionally, in complex multi-step reasoning, and unknown settings (such as novel tool usage or novel environments), even the larger models will still remain uncertain over the best next actions/tokens, and vary considerable over generation steps (see e.g. [2], finding high entropy points in large Qwen models). As a result, we expect entropy-based allocation to continue providing a (model agnostic) benefit over fixed-width search, albeit with potentially diminishing improvements over the baselines as the models become more confident. We will clarify this point in the revision.
>
> Thank you for the review and suggestions, we hope we have addressed any of your remaining concerns.
>
> [1] https://openreview.net/forum?id=CGLoEvCllI
>
> [2] https://neurips.cc/virtual/2025/loc/san-diego/poster/115123

---

> > ### Author Rebuttal · Reviewer_KMmU · 2026-04-01
> >
> > We thank the authors for their thoughtful and thorough rebuttal. We are satisfied with the responses to most of our concerns and are happy to maintain our score.
> >
> > **Conflation of accuracy and efficiency:** We appreciate the authors' commitment to clarifying this distinction in the revision.
> >
> > **Task difficulty:** The use of greedy accuracy as a model-relative proxy for difficulty is reasonable, and we are satisfied with the plan to make this explicit.
> >
> > **Larger models:** The references to work showing non-trivial entropy at 70B scale and high-entropy regions in large Qwen models are helpful and partially alleviate our concern. The honest acknowledgment that gains may diminish with scale is appreciated.
> >
> > **GPU suitability:** We would like to push back gently on this point. The suggestion to treat EDEN as a fixed $B_{\text{max}}$-width beam search with masking does resolve the GPU utilization issue, but at the cost of the core efficiency claim: if $B_{\text{max}}$ forward passes are computed at every step regardless, the expansion savings reported in Table 1 do not translate to computational savings, and EDEN becomes equivalent in cost to beam search with width $b = B_{\text{max}}$. Conversely, if the irregular branching is exploited to actually skip unnecessary forward passes, the GPU batching inefficiency remains.
> >
> > To be clear, we do not think it is the authors' responsibility to design and implement an optimized dynamic batching algorithm; such an algorithm could require extensive low-level systems engineering beyond the scope of this work. It is entirely possible that an efficient implementation exists that captures most of the expansion savings while maintaining reasonable GPU utilization. However, as things stand, the practical efficiency story has a gap: the paper demonstrates fewer expansions, but it remains unclear whether this translates to real-world speedups. We would encourage the authors to acknowledge this tension explicitly in the revision, rather than implying equivalence with beam search in GPU suitability.
> >
> > This concern notwithstanding, we believe this is a solid contribution and are happy to maintain our overall recommendation.

---

> > > ### Author Response · Authors · 2026-04-06
> > >
> > > Thank you for the in-depth follow-up. We appreciate the detailed feedback and are glad that the other points were satisfactorily addressed.
> > >
> > > We agree with the reviewer that there is an inherent tension between exploiting EDEN’s adaptive branching to reduce forward passes and maintaining GPU efficiency via fixed-width batching. In particular, we acknowledge that treating EDEN as a fixed $B_{\max}$-width beam with masking may diminish the practical GPU savings, while exploiting irregular branching can introduce batching inefficiencies.
> > >
> > > We will revise the manuscript to explicitly acknowledge this tradeoff, clarify that the reductions reported in Table 1 correspond to algorithmic expansions, and note that realized wall-clock speedups depend on implementation and system-level considerations. Our intention is to position EDEN as improving the allocation of computation (i.e., where computation is spent), rather than guaranteeing proportional runtime gains, and we will make this distinction explicit in the revision.
> > >
> > > Thank you again for the helpful and constructive suggestions.

---

### Official Review · Reviewer_s2TG · 2026-03-12

**Soundness:** 2
**Presentation:** 2
**Significance:** 3
**Originality:** 3
**Overall Recommendation:** 4
**Confidence:** 3

**Summary:**

This paper proposes EDEN, an entropy-informed decoding method that adaptively adjusts branching factor based on token-level uncertainty. The core idea is simple: allocate more search compute to high-entropy steps and behave more greedily on low-entropy steps. The authors attempt to focus on the aspect of adaptive compute allocation at decoding time, and the authors attempt to analyze a pertinent problem: fixed-width search and standard sampling are both inefficient because they do not adapt to local uncertainty. The method is plug-and-play, does not require retraining, and is supported by both theoretical analysis and experiments on math, code, and science QA benchmarks. The reported results suggest better accuracy-efficiency trade-offs than standard sampling and often competitive performance with beam search at lower expansion cost.

**Compliance With Llm Reviewing Policy:**

Affirmed.

**Final Justification:**

Thank you for your reply, and I am inclined to raise my score.

**Key Questions For Authors:**

How robust is entropy as the control signal when model probabilities are miscalibrated or poorly correlated with actual downstream ambiguity?

How sensitive are results to the specific entropy-to-branching mapping, and would other uncertainty signals perform similarly or better?

Can the authors provide stronger comparisons against more recent adaptive inference or test-time scaling baselines, beyond standard sampling and beam-style methods?

**Limitations:**

ref weakness

**Strengths And Weaknesses:**

Strengths

The paper has a clear and intuitive main idea, with a practical plug-and-play design that does not require model modification or retraining.

A notable strength is the combination of empirical results and theoretical motivation; the paper does more than present a heuristic and attempts to justify why entropy-based adaptive allocation should outperform fixed branching.

The experimental results are reasonably broad and support the main claim that EDEN improves the accuracy-expansion trade-off over standard decoding baselines.

Weekness

The empirical evaluation is still limited to mostly verifiable tasks, and it remains unclear whether the method helps in more open-ended generation settings.

The chosen benchmarks also seem relatively easy. The paper does not include results on harder reasoning benchmarks such as AIME or HLE, so the scalability of the method to more challenging settings is not yet demonstrated.

The theoretical motivation is interesting but somewhat idealized, and the reliability of entropy as a difficulty signal is not fully validated.

---

> ### Author Rebuttal · Authors · 2026-03-30
>
> We thank the reviewer for their comments and suggestions. We are happy the reviewer finds the main idea to be clear, intuitive, and practial, and likes the empirical results and theoretical motivations.
>
> We hope the points below address any of the reviewers' remaining concerns and key questions:
>
> * **Verifiable tasks**: We agree that due to the scoring selection, these approaches (and beam search in general) are more suited to verifiable tasks rather than open-ended generation. This is inherited from the underlying tree search approaches, and not something we address in this paper or suggest the method to be used for. For this reason, we have kept it out of focus, and mentioned as a limitation in Appendix F. However, in Appendix E.4. we do show cases of using different style scoring algorithms, where these could be catered towards open-ended generation, even if its not a *verification* based scorer. We will make this clearer in the main body.
>
> * **Benchmarks**: As suggested, we have now added results on the more complex domains of AIME (both 2024 and 2025) and HLE (subset of 100 text only MCQs):
>
> ||AIME24|AIME25|HLE|
> |-|-|-|-|
> |Greedy |3.3\%|0.0\%|5.0\%|
> |Top-$k| 0.0\%|0.0\%|3.0\%|
> |Top-$p$| 3.3\%|0.0\%|5.0\%|
> |Min-$p$| 0.0\%|0.0\%|4.0\%|
> |Best-of-$n$|0.0\%|0.0\%|6.0\%|
> |Majority Voting|3.3\%|**6.7\%**|4.0\%|
> |Beam Search |3.3\%|0.0\%|5.0\%|
> |**Ours (EDEN)**|**10\%**|**6.7\%**|**8.0\%**|
>
>
> where EDEN is still able to result in an improvement, even when the base model gets near 0 questions correct. On AIME24, we see three questions are answered correctly, a large (relative) improvement beyond the comparisons which only get at most one correct. Likewise on AIME25, the only two approaches which get corect answers are majority voting and the proposed, further strengthening the support for EDEN on additional complex datasets (in addition to Table 1). For HLE, EDEN outperforms all comparisons. These result show the applicability of EDEN to more challenging settings, even those where the base model struggles significantly, where the proposed approach continues to achieve the highest accuracy and provide improvements to the base model.
>
> * **Idealized setting**: While the theoretical motivation does rely on certain assumptions of the model, in the Appendix we empirically justify these assumptions with real model usage (Fig 9 and Fig 10), showing these to hold. We hope this emperical validation, in combination with the theory, helps validate the usage of entropy as a difficulty signal.
>
> * **Entropy**:
>     * Miscalibration: We have targeted this question in Appendix D.2, showing the approach still performs well under miscalibration, allowing errors upto $0.5/B_{\max}$ without changing branching decisions.
>     * Sensitivity: We address sensitivity to entropy estimation under limited sampling in Fig 6, to paraphrasing in Fig 9, and to miscalibration in Appendix D.2. Across these settings, EDEN remains stable, suggesting that it is not overly brittle to small perturbations in the uncertainty signal.  Importantly, the method is not tied to a specific entropy-to-branching rule. As discussed in Section 3.1.2, the theory permits any mapping in which branching is monotone in entropy; the linear rule we use is a simple default rather than a uniquely optimal choice, and in response to reviewer gouc we have added an additional experiment on this with a nonlinear mapping, demonstrating further improved performance.
>     * We also point the reviewer to our discussion with reviewer pgbz where we perform additional experiments to validate the sensitivity to miscalibration.
>
> * **Other uncertainty signals**: Our use of entropy is theoretically motivated (Section 3.1.1): entropy captures both the number of plausible next-token candidates and the difficulty of selecting among them, which directly motivates allocating more compute to higher-entropy steps. In this sense, entropy is the uncertainty signal most closely aligned with our theoretical framework. That said, we do not claim that entropy is uniquely optimal. Other uncertainty signals (e.g., margin-based) may also work well. We view investigating these alternative signals an important direction for future work, with this work focusing on entropy specifically as the signal.
>
> * **Baselines**: We added an extra comparison to Diverse tree search/diverse group beam search, another widely used test-time scaling baseline, and the proposed approach strongly outperformed this ($P($ EDEN $>$ Diverse Tree Search$) = 0.96$):
>
> ||GSM8K|Math-500|HumanEval|SciBench|
> |-|-|-|-|-|
> | Diverse Tree Search|77.5%±2.2%|30.0%±4.0%|26.4±6.7%|5.4%±1.7%|
> |**EDEN**| **80.5%±2.2%**|**32.8%±4.0%**|**31.3±7.4%**|**7.4%±2.0%**|
>
> Thank you for the suggestions. We hope the extensions and answers will be sufficient to address your questions and concerns.

---

> > ### Author Rebuttal · Reviewer_s2TG · 2026-04-01
> >
> > From the results, current models still achieve relatively low scores on benchmarks such as AIME. Could you further verify whether the method remains effective on larger models?

---

> > > ### Author Response · Authors · 2026-04-04
> > >
> > > We thank the reviewer for their timely acknowledgement, and are glad the original rebuttal addressed the other concerns. Below we focus on the persisting concern of the *AIME* accuracy and larger models.
> > >
> > > **AIME** We would like to stress that the reported AIME results are inline with existing models upto 9B in size (including those specifically trained for math), as seen for example at: https://huggingface.co/mistralai/Mathstral-7B-v0.1 which reports similar results on larger models, with the best getting 2/30. Where above, we show 3/30 correct.  This shows the proposed approach can achieve competitive performance with larger models using a smaller model with more targeted decoding, a useful result in itself.
> > >
> > > However, to further address the concerns raised on both the AIME accuracy, and the larger model sizes, we run new experiments on the brand new  state-of-the-art *Gemma4 model* (released during the rebuttal period) with 8B parameters (4 million effective) on AIME24. From this, we observe strong performance across decoding methods, with EDEN again achieving the highest accuracy:
> > >
> > >
> > > | | Accuracy |
> > > | -| - |
> > > | Greedy     |40.0\%   |
> > > | Top-$k     |36.7\% |
> > > | Top-$p$     |40.0\%   |
> > > | Min-$p$     |40.0\%   |
> > > | Best-of-$n$     |40.0\%  |
> > > | Majority Voting     |40.0\%  |
> > > | Beam Search     |40.0\%  |
> > > | **Ours (EDEN)**   | **43.3\%** |
> > >
> > >
> > > These results demonstrate consistent improvements on larger models, with EDEN achieving the highest accuracy across the decoding methods.
> > >
> > > **Larger models**: We confirm that EDEN remains effective on larger models, as demonstrated by the new results on Gemma4 (8B) and the additional experiments on Mistral (7B) (Appendix B.1.1), where EDEN consistently improves over standard decoding methods.
> > >
> > > While our experiments focus on models up to 8B parameters, we observe consistent improvements across all evaluated scales, including the newly added results above, and these findings are supported by theoretical guarantees that extend beyond the tested regime.
> > >
> > > **Theory**: To provide further validation to the theoretical guarantees holding in larger settings (e.g. beyond 8B), recent work [1] demonstrates that the entropy in the output distributions remain in models up to 70B at least (the largest evaluated), and that reductions in output entropy scales very *slowly* with model size. Likewise, in large models (Qwen 32B), these high entropy points remain and become essential tokens driving the resulting generations [2]. This means there continues to be room for targeted improvement from the proposed approach, even with the larger models ($\gg$ 8B). While this room for improvement may reduce as the models become larger, [1] shows this should be a relatively small reduction, and [2] shows these high entropy points to be impactful even in larger models.
> > >
> > > Across six datasets (GSM8K, MATH500, HumanEval, SciBench, AIME, HLE), four model families (Llama, Granite, Gemma, Mistral), and scales from 1B to 8B parameters, EDEN consistently improves over standard decoding methods, with gains persisting on the largest evaluated models (Gemma4-8B, Mistral-7B).
> > >
> > > We hope these additional results and analyses address the remaining concerns, further supporting the effectiveness of EDEN on larger models and more complex datasets. Thank you again for the review and prompt rebuttal follow-up.
> > >
> > >
> > > [1] https://neurips.cc/virtual/2025/loc/san-diego/poster/119303
> > >
> > > [2] https://neurips.cc/virtual/2025/loc/san-diego/poster/115123

---

### Official Review · Reviewer_gouc · 2026-03-13

**Soundness:** 3
**Presentation:** 2
**Significance:** 2
**Originality:** 2
**Overall Recommendation:** 4
**Confidence:** 3

**Summary:**

This paper introduces EDEN, a decoding method that adptively branches beam search based on next-token entropy, allowing for more search in uncertain regions, and behaving more greedily when the model is confident. Theoretical analysis show that this entropy-monotone allocation is strictly better than fixed-width policies. Experiments are conducted on math and science reasoning tasks, showing superior performance compared to standard sampling/search baselines.

**Compliance With Llm Reviewing Policy:**

Affirmed.

**Final Justification:**

The authors have addressed my concerns with convincing explanations and experiments in the rebuttal.

In terms of the open-ended generation tasks, I recommend the authors add experiments to compare with other beam-search-like methods, and conduct corresponding analysis, even if EDEN may not be state-of-the-art on open-ended generation tasks.

Overall, the authors' responses have strengthened their claim, so I have raised my score to 4.

**Key Questions For Authors:**

1. How would the authors suggest to set $B_{\max}$ for different tasks/models?

2. Why are the main results measured in PASS@1 now that beam search methods are used to generate multiple candidates?

**Limitations:**

Yes.

**Strengths And Weaknesses:**

- Strengths

1. The motivation is reasonable. Fixed-width beam search may lack exploration into uncertain tokens, while wasting computation on confident tokens.

2. The authors also provide theoretical justification for this motivation, showing that entropy-monotone allocation is strictly better than fixed-width search.

3. The experiments verify that EDEN is indeed more efficient than fixed-width beam search with fewer expansions.

- Weaknesses

1. The branching design $B_t=max(1, \lfloor B_{\max}\ \cdot \bar H_t\rfloor)$ is relatively intuitive. Although the theory supports monotone entropy-based allocation, in practice, the authors use a linear function to integrate entropy.

2. The paper does not evaluate on open-ended generation datasets, where beam-search-like methods often don't perform well. Such tasks are further emphasized since the proposed entropy method depends on the token distribution, and does not rely on semantic information.

3. Although discussed in the related work, beam-search-like / entropy-based adaptive methods are not compared as baselines.

4. As the authors observe, different dataset difficulties can lead to different expansions, and expansions may be affected more factors. This suggests that the hyperparameter $B_{\max}$ may not be consistent across tasks/models, making it hard to tune. Additionally, for users of standard beam search methods, relevant hyperparameter knowledge may also not transfer directly.

---

> ### Author Rebuttal · Authors · 2026-03-30
>
> Thank you for the helpful review. We are glad you recognise the approach's motivation and theoretical grounding.
>
> We focus on the remaining questions and weaknesses mentioned:
>
> * *“The branching design  is relatively intuitive”*:  We agree the specific linear form used $B_t = \max(1, \lfloor B_{\max} * H_t \rfloor)$ is simple, while the theory itself is more general. This is intentional: our contribution is the *principle* of entropy-monotone allocation, so we favored a simple implementation to demonstrate the benefit, which outperformed the existing baselines without adding additional complexity. To evaluate the robustness to more complex forms, we have now performed additional ablations with a nonlinear mapping $B_t = \min(B_{\max}, e^{H_t})$ on a random subset of examples:
>
> ||GSM8K|MATH500|HumanEval|
> |-|-|-|-|
> |EDEN (Linear)|80.0%|31.0%|27.6%|
> |EDEN (Nonlinear)|**84.0%**|**35.0%**|27.0%|
>
> showing the non-linear version also performs well, and *can potentially further* improve performance. This suggests that the improvements stem from the underlying monotone-in-entropy allocation principle proposed, rather than the specific functional form, with additional room to improve upon the linear rule.
>
> * **Open-ended generation**: We agree that beam-style methods, including EDEN, are less suitable for open-ended generation (e.g., creative writing), where likelihood may not align with quality. We will clarify this limitation in the main text (previously in Appendix F). Additionally, in Appendix E.4 we discuss the potential for alternative scorers, which may be more useful for open ended generation, e.g. with diversity awarding scorers.
>
> * **Semantic information**: We acknowledge that EDEN’s reliance on token-level entropy may not fully capture semantic uncertainty. However, the EDEN framework is not restricted to token entropy: the theoretical result applies to other entropy-based measures, such as semantic entropy [1]. This fits naturally within EDEN and enables grouping at the semantic level without changing the underlying allocation principle. This suggests EDEN could in theory operate at different abstraction levels (token vs semantic) without changing the framework, just the node selection/expansion style.
>
> * **Comparisons**: We have added an extra beam search-based baseline (diverse beam search), and an additional entropy-based baseline top-$h$ [3], **in addition** to the existing beam search, best-of-n, and majority voting comparisons. We are including the new baselines below:
>
> ||GSM8K|MATH500|HumanEval|SciBench|
> |-|-|-|-|-|
> |top-$h$|69.7%±2.5%|26.0%±3.8%|27.0%±6.7%|4.5%±1.5%|
> |Diverse Tree Search|77.5%±2.2%|30.0%±4.0%|26.4%±6.7%| 5.4%±1.7%|
> |**EDEN** (Ours)|**80.5%±2.2%**|**32.8%±4.0%**|**31.3%±7.4%**|**7.4%±2.0%**|
>
> demonstrating that the proposed approach EDEN also outperforms diverse tree search and entropy comparisons, with posterior probability $P($EDEN $>$ Diverse Tree Search$) = 0.96$, and $P($EDEN $>$ top-$h)=0.99$, in addition to already outperforming beam search, best of-$n$ and majority voting (Table 1).
>
> * **Hyperparameters**: We agree hyperparameter robustness is important. In EDEN, $B_{\max}$ plays the same role as beam width, but is *less sensitive*, since branching adapts below this maximum. Empirically, performance is monotonic in $B_{\max}$ (Fig 4). In practice, we recommend setting $B_{\max}$ based on compute budget (e.g., equivalent to beam width), after which EDEN adapts automatically without further tuning, unlike methods that require per-task calibration to not waste compute on simpler problems. We emphasize that variation in expansions across tasks (Fig 5) requires no additional tuning. Given a fixed $B_{\max}$, EDEN automatically adapts branching based on entropy, allocating more compute to uncertain steps and remaining efficient on simpler ones.
>
> * **PASS@1**: While EDEN generates multiple candidates, its objective is to identify a *single* trajectory, rather than to maximize the probability that any of a set of trajectories is correct, making pass@1 the most directly aligned metric (and also breaching the independence assumption required for pass@k, due to the shared prefixes in the tree). Nevertheless, for completeness we report: pass@k for EDEN on a random subset of MATH500:
>
> ||pass@1|pass@3|pass@5|
> |-|-|-|-|
> |EDEN|30.9%±4.5%|35.1%±4.8%|36.6%±4.9%|
>
> demonstrating that considering the top $k$ candidates from the resulting tree *can* further improve performance, but note that this was not explictly selected/optimised for.  If pass@k is the goal, then the scoring criterion should be adjusted to be over a set of trajectories, in the spirit of [2], rather than ranking paths in isolation.
>
> We thank the reviewer again for the feedback and hope the additional experimentation and clarifications address any remaining concerns.
>
> [1] https://www.nature.com/articles/s41586-024-07421-0
>
> [2] https://arxiv.org/abs/2505.15201
>
> [3] https://neurips.cc/virtual/2025/loc/san-diego/poster/120050

---

> > ### Author Rebuttal · Reviewer_gouc · 2026-04-02
> >
> > The authors have addressed my concerns with convincing explanations and experiments.
> >
> > In terms of the open-ended generation tasks, I recommend the authors add experiments to compare with other beam-search-like methods, and conduct corresponding analysis, even if EDEN may not be state-of-the-art on open-ended generation tasks.
> >
> > Overall, the authors' responses have strengthened their claim, so I have raised my score to 4.

---

> > > ### Author Response · Authors · 2026-04-04
> > >
> > > We thank the reviewer for their prompt response, and we are glad we have addressed their other concerns. We appreciate the in-depth initial review and follow-up. Below, we look at the remaining suggestion.
> > >
> > >
> > > **Open-ended generation** We have followed your recommendation, and performed additional experimentation on open-ended tasks. Specifically, we have used the OpenMeva benchmark [1] for open-ended generation and selected a random subset of 100 testing points (from the WritingPoints dataset) for evaluation.
> > >
> > > We evaluate open-ended generation using four complementary metrics:
> > > - (a) Semantic quality: cosine similarity to a human reference (evaluation only) using a sentence-transformers model (all-MiniLM-L6-v2);
> > > - (b) Logical consistency: contradiction rate between adjacent sentences using DeBERTa-MNLI;
> > > - (c) Repetition: percentage of repeated n-grams;
> > > - (d) Relatedness: similarity between each sentence and the centroid of the remaining sentences.
> > >
> > > Metrics (a) and (d) should be maximized, while (b) and (c) should be minimized. This multi-metric evaluation mitigates the limitations of any single metric as discussed in [1].
> > >
> > > The resulting metrics are:
> > >
> > >
> > > | | Semantic Quality ↑ | Contradiction Rate ↓ | Repetition ↓ | Relatedness ↑ |
> > > | -| - | - |  - |  -|
> > > | Greedy     |0.381 | 3.5\% | 3.8\% |37.1\% |
> > > | Top-$k     |0.382 | 4.5\% | 3.4\% | 34.3\% |
> > > | Top-$p$     | 0.381 | 3.6\% | 3.8\% | 37.1\% |
> > > | Min-$p$     | 0.376 | 4.0\% | 4.0\% |36.5\% |
> > > | Best-of-$n$     | 0.392 | 4.3\% |  **1.8\%** | 34.7\% |
> > > | Majority Voting     |0.381 | 3.8\% | **1.8\%** | 33.3\% |
> > > | Beam Search     | 0.397 | **2.8\%** | 4.0\% | 36.3\% |
> > > | **Ours (EDEN)**   | **0.408**| 3.3\% (2nd) | 4.2\% | **37.2\%** |
> > >
> > >
> > > EDEN achieves strong performance in semantic quality, contradiction rate, and intra-text relatedness, with the highest cosine similarity, a low contradiction rate, and the best relatedness score. These improvements stem from EDEN’s entropy-guided branching, which concentrates exploration in uncertain regions while remaining selective in confident ones, resulting in more globally consistent and semantically coherent generations.
> > >
> > > However, consistent with prior work on search-based decoding [2], EDEN (and beam search) exhibits higher repetition than sampling-based methods, reflecting a known trade-off between coherence and diversity.
> > >
> > > To combat this repetition in the proposed approach, we can utilise the reward model formulation discussed in Appendix B.2. By adding a small bonus reward for minimising repetition, i.e.,
> > >
> > > $$
> > > \rho_t = -\frac{
> > > \sum_{g \in G_n(y_{1:t})}
> > > \max(0, C(g) - 1)
> > > }{
> > > \lvert G_n(y_{1:t}) \rvert
> > > }
> > > $$
> > >
> > > where $\rho_t$ corresponds to the negative n-gram repetition rate of a sequence $y_{1:t}$, $G$ denotes the multiset of all contiguous $n$-grams in the sequence, and $C(g)$ is the number of occurrences of $n$-gram $g$. Through  adjusting the reward strength  $\lambda$ (see Appendix B.2 for definition), the resulting repetition rates can be controlled:
> > >
> > > | | Repetition Rate |
> > > | -------- | -------- |
> > > | EDEN  $\lambda=0.00$    | 4.2\%     |
> > > | EDEN  $\lambda=0.25$    | 3.7\%     |
> > > | EDEN  $\lambda=1.0$    | 3.3\%     |
> > >
> > > showing how custom reward scorers can be used under the proposed approach to further improve open-ended generation, and guide the decoding towards more desirable outcomes.
> > >
> > >
> > > We think this is a great direction to follow, and are thankful for the suggestion. This gives an additional dimension and application area for the proposed approach. We will include the above results and discussion in the revised manuscript. We hope this additional experiment addresses any remaining concerns. Once again, we thank the reviewer for the great suggestion.
> > >
> > >
> > > [1] https://aclanthology.org/2021.acl-long.500/
> > >
> > > [2] https://arxiv.org/abs/2005.11009

---

### Official Review · Reviewer_Pgbz · 2026-03-16

**Soundness:** 3
**Presentation:** 2
**Significance:** 2
**Originality:** 3
**Overall Recommendation:** 4
**Confidence:** 3

**Summary:**

The paper proposes EDEN (Entropy-informed Decoding) as a principled way for adaptive resource allocation during LLM decoding. Rather than using a fixed beam width, EDEN adjusts the branching factor at each decoding step based on the entropy of the next-token distribution, pruning candidates whose score upper bounds fall below the current best. The main reason for this is that high-entropy tokens require more exploration. The authors prove that branching factors monotone in entropy improve on fixed-width beam search (in terms of regret), and provide sampling bounds for entropy estimation from limited API access (e.g., top-$k$ log-probs).

**Compliance With Llm Reviewing Policy:**

Affirmed.

**Final Justification:**

The rebuttal addressed my main concerns. The authors provide a regret bound and explanation for why EDEN is robust to entropy estimation.

**Key Questions For Authors:**

1. How is entropy computed in practice for closed-source models that expose only top-$k$ log-probs? The entropy over a truncated distribution is a biased estimate of the true entropy. How sensitive is EDEN to this bias? Have you measured the correlation between top-$k$ entropy and full-vocabulary entropy across different domains and model sizes?
 2.  Theorem 3.4 mentions that entropy varies across decoding steps for most LLM generation processes, but the paper does not empirically characterize this variation for real LLM generations. Can you quantify how task-dependent this variation is?
 3. Can you give a concrete bound on $R_T$ as a function of $T$? Theorem E.2 seems to establish dominance but not the rate.

**Limitations:**

The statistical reliability of the results is the main concern. The high variance and overlapping confidence intervals make it difficult to draw conclusions about EDEN's reliability. The limited range of tested models leaves open questions about performance in larger models where entropy profiles may differ substantially. Additionally, the paper does not provide an explicit regret bound on $R_T$.

**Strengths And Weaknesses:**

## Strengths

The adaptive branching strategy is well motivated by information theoretic arguments. Proposition 3.3 shows that the optimal budget scales with entropy, and Theorem E.2 shows that entropy-based policy dominates fixed-allocation policies. The method can be used on closed-source APIs and it achieves comparable or better accuracy to beam search with fewer token expansions, e.g., matching Beam-3 on GSM8K with fewer expansions.

## Weaknesses

 The notations, for example $V_t$ in Line 149, should be defined earlier in the paper and in general, the writing should be improved. The main experiments use only Llama-3.2-3B-Instruct and appendix results are on Gemma-2-2b-it, Granite-3.3-2b, and Mistral-7B. However, no large-scale models are benchmarked. It is unclear whether the entropy-based allocation remains beneficial when the base model is already highly capable. The reported 95% bootstrapped confidence intervals are large and frequently overlap between EDEN and baselines. For example, on GSM8K, EDEN achieves 80.5% $\pm$ 2.2% vs. Beam-3 at 81.5% $\pm$ 2.2%. On HumanEval: 31.3% $\pm$ 7.4% vs.\ Greedy 27.0% $\pm$ 7.1%. This undermines the statistical significance of the improvements.
Given the above variance, it is unclear how to make the method reliably outperform baselines. The paper also mentions cumulative regret in Theorem 3.4 and proves dominance over fixed policies, but does not provide an explicit regret bound.

---

> ### Author Rebuttal · Authors · 2026-03-26
>
> We thank the reviewer for their review and the strengths highlighted. We are glad the reviewer finds the method to be well motivated by information-theoretic arguments and useful for both open models and closed-source APIs.
>
> Below, we focus on addressing the remaining limitations outlined:
>
> * **Statistical reliability**: While per-dataset confidence intervals can overlap, the overall pattern is consistent: EDEN is competitive with or better than the strongest baselines on every dataset, and the cross-dataset Bayesian hierarchical analysis (as suggested in [1,2]) aggregating this across dataset information shows a 75% posterior probability that EDEN is the best method overall, with pairwise win probability at least 96% against most baselines and 77% against beam search (Table 1 and Fig 3), indicating a consistent improvement *across* tasks irrespective of the *within* task variation.
>
> * **Concrete bound on $R_T$**: This is a great point, and we have now derived an explicit bound on cumulative regret as a function of $T$. Under a standard gap assumption, $R_T$ decays exponentially in the per-step budget $m_t$:
> $$\mathbb{E}[R_T] \le C \sum_{t=1}^T \exp \big(-c m_t \Delta_{\min}^2\big)$$ for constants $C,c > 0$.
> yielding sublinear or even bounded regret under logarithmic allocation schedules (key result only due to space constraints, glad to provide full steps in follow up response).
> * **Closed-source entropy**: This is correct: in the closed-source setting we compute entropy over the top-k log-probabilities (i.e., the renormalized truncated distribution), which introduces a downward bias due to omitted tail mass. However, EDEN does not require an unbiased estimate, only a stable proxy. As shown in Appendix D.2, branching decisions are invariant whenever $|H - \hat{H}| < 0.5 / B_{\max}$ making the method robust to moderate estimation error.  Empirically, Fig 6 shows entropy noise remains well within this regime, and Fig 7 shows  performance differences between full-logit and top-k entropy are negligible, indicating that EDEN is insensitive to the bias introduced by this truncation. We ran additional experiments across the datasets and models demonstrating the correlation (which we will add to the appendix), including a subset here for your reference (on Llama):
>
> | Domain | k | Spearman | Pearson | Safe $B_{\max}$ (median) |
> |-|-:|-:|-:|-:|
> | gsm8k | 5 | 0.77 | 0.79 | 6 |
> |  | 10 | 0.90 | 0.89 | 8 |
> |  | 20 | 0.95 | 0.95 | 12 |
> | math500 | 5 | 0.81 | 0.86 | 3 |
> |  | 10 | 0.89 | 0.92 | 4 |
> |  | 20 | 0.94 | 0.95 | 6 |
> | humaneval | 5 | 0.99 | 0.99 | 50 |
> |  | 10 | 0.99 | 0.99 | 50 |
> |  | 20 | 0.99 | 0.99 | 50 |
> | scibench | 5 | 0.75 | 0.83 | 2 |
> |  | 10 | 0.84 | 0.90 | 3 |
> |  | 20 | 0.90 | 0.94 | 5 |
>
> showing strong alignment between top-k and full-logit $H$, even with low $k$. We report the safe branching factor (e.g. where the estimation error does not alter branching factor, capped at 50) in the rightmost column, which paired with Fig 7 suggests truncated entropy preserves the relevant signal, consistent with other recent work [3].
>
> * **Entropy across steps**: This is a good point. The key assumption in Theorem 3.4 is that entropy varies across decoding steps, i.e., $Var(H_t) >0$, motivating heterogeneous $b_t$ (to obtain the benefits seen in Fig 2). Empirically, we already observe this indirectly in Fig 5, where EDEN allocates different numbers of expansions across tasks despite using the same model and decoding rule, reflecting systematic variation in entropy across both steps and domains. To make this explicit, we ran additional experiments tracking $H_t$, demonstrating substantial variation, i.e. (for Llama):
> |  | Var(Hₜ) |
> |-|-:|
> | gsm8k | 0.49 |
> | math500 | 0.58 |
> | humaneval | 0.69 |
> | scibench | 1.04 |
>
> confirming  $Var(H_t) \gg 0$ in practice, motivating the adaptive branching, and also indicating that the magnitude of variation is task-dependent. The empirically observed variability motivates the theoretical basis for entropy-adaptive allocation from Theorem 3.4, and explains why EDEN outperforms fixed branching strategies due to the dynamic branching factor.
>
> * **Performance on larger models**: We agree this is important. As a preliminary check, we include results on Mistral-7B (Appendix B.1.1), where EDEN continues to match or outperform baselines under the same decoding setup. While we have not yet evaluated very large (>10B) models due to compute constraints, please see discussion with reviewer KMmU about the potential.
> * **Notation**: We will update the manuscript to be clearer about the notation (following Table 3 in the appendix).
>
> We hope the above points, particularly the addition of regret bounds, clarification on the statistical analysis, and additional justification and experimentation on entropy, will help address the reviewers' concerns.
>
> [1] https://jmlr.org/papers/v18/16-305.html
>
> [2] https://link.springer.com/article/10.1007/s10994-017-5641-9
>
> [3] https://arxiv.org/abs/2603.03310

---

> > ### Author Rebuttal · Reviewer_Pgbz · 2026-04-04
> >
> > I thank the authors for the explanations. I will increase my score.

---

### Decision · Program_Chairs · 2026-04-30

**Decision:**

Accept (regular)

**Comment:**

All authors feel that their core concerns were addressed and agree that this is a theoretically motivated, interesting paper.  One reviewer specifically clarifies that they expect the authors to frame the contribution in a way that does not exaggerate wall clock improvements, but still believes that the ideas are sound and interesting.